# Dynamics-Informed Protein Design with Structure Conditioning

**Urszula Julia Komorowska,**[*] **Simon V Mathis,**[*] **Kieran Didi, Francisco Vargas,**
**Pietro Lio & Mateja Jamnik**
Department of Computer Science and Technology
University of Cambridge
Cambridge, CB30FD, UK
`{ujk21, svm34, ked48, fav25, pl219, mj201}@cam.ac.uk`

## Abstract

Current protein generative models are able to design novel backbones with desired shapes or functional motifs. However, despite the importance of a protein's dynamical properties for its function, conditioning on these dynamics remains elusive. We present a new approach to include dynamical properties in protein generative modeling by leveraging Normal Mode Analysis. We introduce a method for conditioning diffusion probabilistic models on protein dynamics, specifically on the lowest non-trivial normal mode of oscillation. Our method, similar to classifier guidance conditioning, formulates the sampling process as being driven by conditional and unconditional terms. However, unlike previous works, we approximate the conditional term with a simple analytical function rather than an external neural network, thus making the eigenvector calculations approachable. We present the corresponding SDE theory as a formal justification of our approach. We extend our framework to conditioning on structure *and* dynamics at the same time, enabling scaffolding of dynamical motifs. We demonstrate the empirical effectiveness of our method by turning the open-source unconditional protein diffusion model Genie into a normal-mode-dynamics-conditional model with no retraining. Generated proteins exhibit the desired dynamical and structural properties while still being biologically plausible. Our work represents a first step towards incorporating dynamical behaviour in protein design and may open the door to designing more flexible and functional proteins in the future.

## 1 Introduction

Generative Artificial Intelligence (AI) has rapidly accelerated protein design research. A common problem tackled with AI is the task of protein backbone design, which is finding a new and realistic 3D structure tailored to the specific biological function. Recently, AI models based on the denoising diffusion framework (Ho et al., 2020; Song et al., 2021) have shown remarkable success in generating realistic protein backbones, especially backbones with pre-defined, fixed substructures often referred to as *motifs* (Watson et al., 2022; Trippe et al., 2023). Since many functions have been linked to the presence of various functional motifs, enforcing the generation process to preserve such substructures is crucial in meaningful protein design. However, current modeling approaches do not incorporate an important aspect of protein design - structure alone is not enough to determine the protein's functional properties. Information about protein flexibility, especially about its low-frequency collective motion, is crucial in determining protein functional properties (Bauer et al., 2019). In this work, we address this research gap and provide a framework for a diffusion model conditioned not only on structural constraints but also on protein dynamics.

We analyse protein dynamics through the lens of Normal Mode Analysis (NMA) (Bahar et al., 2010). This is a simple yet powerful method for obtaining eigenvectors of the motion of protein residues and their relative displacements in each mode. After performing NMA on a real-life protein with known functionality, the obtained eigenvectors can be used as the dynamic targets when using a diffusion

---

[*]Equal contributions.

model to sample a novel backbone. We are particularly interested in proteins which exhibit hinge-like motions, which are responsible for a number of protein functions and are strongly constrained in both structure and dynamics (Khade et al., 2020). Protein hinges usually involve two secondary structure elements rotating against each other about the common axis, similar to how a hinge at the door frame has closing and opening motions.

Our contributions are as follows:

- We introduce a new methodology for conditioning protein generation on dynamical properties. Our approach is based on NMA which is easy to compute and captures collective motions related to protein function. Moreover, we demonstrate how conditioning on the desired relative displacements, which we refer to as *dynamics conditioning*, can be accompanied by *structure conditioning*. To substantiate this *joint conditioning* theoretically, we present a formal interpretation in terms of stochastic differential equations.

- We train our custom conditional diffusion model and generate dynamics-conditioned backbones. Thanks to the large number of real-life dynamics targets extracted from our data, we provide a detailed analysis of the effectiveness of the method. We measure the agreement of the displacements using a custom loss function and manually inspect the agreement of target and sample displacement vectors for selected samples. Our method indeed allows us to generate proteins with desired dynamics and is easily transferable to other models.

- We showcase the joint conditioning by applying it to a trained Genie model (Lin & AlQuraishi, 2023). Through literature research, we select three proteins that exhibit hinge structures and motions, identify residues located in the hinge arms and use those as conditioning targets. Figure 1 shows that we succeed in generating new and biologically plausible proteins with the targeted hinge dynamics, demonstrating that our framework can be transferred to other models in a plug-and-play fashion.

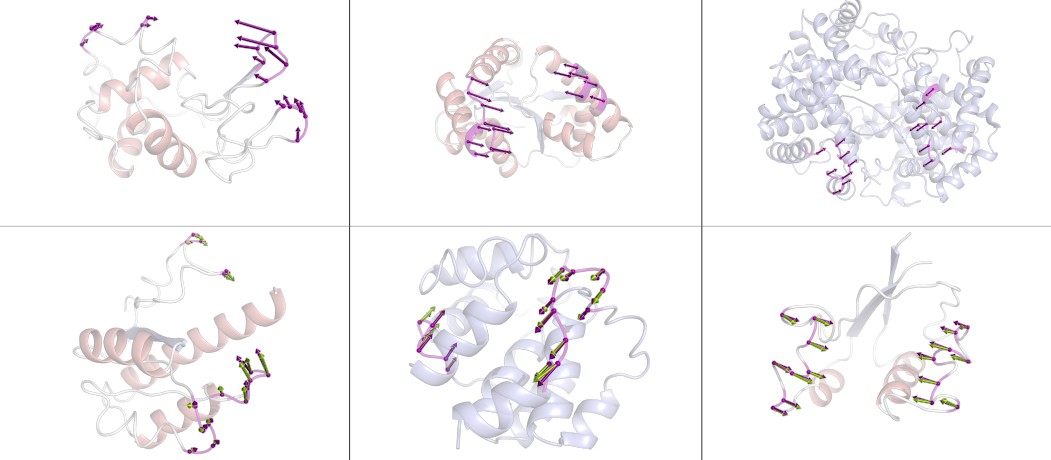

Figure 1: Comparison of natural proteins (top) from which the hinge targets were extracted with conditional samples (bottom). **Top row:** from the left – lysozyme, adenylate kinase, haemoglobin. **Bottom row:** protein backbones synthesised with Genie that match the pre-selected hinge motif residues and have the desired dynamics, from the left with lysozyme, adenylate kinase, haemoglobin targets. Purple arrows are the displacements of selected residues in the normal mode, while green ones are the displacements in the same mode but in a novel structure. Arrows have been scaled up for increased visual clarity. Note how the relative amplitudes and pair-wise angles of the green arrows match the constraints imposed by the target, and how the relative positions of the novel hinge residues are as in the original structure.

## 2 BACKGROUND AND RELATED WORK

### 2.1 DIFFUSION PROBABILISTIC MODELING

The generative process in diffusion probabilistic models (Sohl-Dickstein et al., 2015) starts with a sample from the standard normal distribution, $x_T \sim \mathcal{N}(0, 1)$. The goal of this process is to transform $x_T$ into the sample $x_0$ from the targeted data distribution $p_0(x_0)$, initially unknown and indirectly accessed by the trained model.

The key idea is to formulate the model training as a forward diffusion process in which the model predicts how much noise was added to the original sample. For a sample from the training set $x_0$, the forward process is defined as iteratively adding a small amount of Gaussian noise to the sample in $T$ steps, which produces a sequence of noisy samples $x_{0:T}$ such that the final sample $x_T \sim \mathcal{N}(0, 1)$ to good approximation. In the Denoising Diffusion Probabilistic Modeling (DDPM) framework (Ho et al., 2020) the noise magnitude at each step is defined by a variance schedule $\{\beta_t, t \in [0 : T]\}$ such that

$$p_t(x_t|x_{t-1}) = \mathcal{N}(x_t, \sqrt{1 - \beta_t}x_{t-1}, \beta_t I). \tag{1}$$

The above transition defines a Markov process in which the original data is transformed into a standard normal distribution. It is possible to write the density of $x_t$ given $x_0$ in a closed form

$$p_t(x_t|x_0) = \mathcal{N}(x_t, \sqrt{\bar{\alpha}_t}x_0, (1 - \bar{\alpha}_t)I), \quad \text{s.t.} \quad x_t = \sqrt{\bar{\alpha}_t}x_0 + \sqrt{1 - \bar{\alpha}_t}\epsilon_t, \tag{2}$$

where $\bar{\alpha}_t = \prod_i^t \alpha_i$ and $\alpha_i = 1 - \beta_i$ and $\epsilon_t \sim \mathcal{N}(0, 1)$. Transforming a sample $x_T$ into the sample $x_0$ is done in several updates that reverse the destructive noising, given by a reverse sampling scheme

$$x_{t-1} = \frac{1}{\sqrt{\alpha_t}} \left( x_t - \frac{\sqrt{1 - \alpha_t}}{1 - \bar{\alpha}_t} \epsilon_\theta(x_t, t) \right) + (1 - \alpha_t)z, \tag{3}$$

where $z \sim \mathcal{N}(0, 1)$. The neural network $\epsilon_\theta$ (the denoiser) should be trained to predict noise added to $x_0$. Ho et al. (2020) showed the following loss function is sufficient

$$L = \mathbb{E}_{x_0, t} \left( ||\epsilon_t - \epsilon_\theta(\sqrt{\bar{\alpha}_t}x_0 + \sqrt{1 - \bar{\alpha}_t}\epsilon_t, t)||^2 \right). \tag{4}$$

Song et al. (2021) state that the DDPM is an example from the larger class of score-based models. They demonstrated that the discrete forward and reverse diffusion processes have their continuous time equivalents, that is, the forward Stochastic Differential Equation

$$dx = -\frac{1}{2}\beta(t)xdt + \sqrt{\beta(t)}dw, \tag{5}$$

and its reversal

$$dx = \left[ -\frac{1}{2}\beta(t)x - \beta(t)\nabla_x \ln p_t(x) \right] dt + \sqrt{\beta(t)}d\bar{w}, \tag{6}$$

where the quantity $\nabla_{x_t} \ln p_t(x_t)$ is called the score and is closely related to the noise in DDPM by the equivalence $\nabla_{x_t} \ln p_t(x_t) = -\epsilon_t/\sqrt{1 - \bar{\alpha}_t}$ (derivation are in the Appendix F). Any model trained to predict the noise can be written in terms of the score, which is an essential property of our work. Whenever we derive some expression with respect to the score, we can use the noise-based formulation for forward and reverse diffusion processes by simply substituting $\epsilon_t = -\sqrt{1 - \bar{\alpha}_t}\nabla_{x_t} \ln p_t(x_t)$.

**Related work on Diffusion Probabilistic Models for protein design.** In the context of protein generative modelling, the real data samples $x_0$ are often represented by protein backbone coordinates (e.g., at the resolution of $C_\alpha$ atoms), optionally with amino-acid identity as a scalar feature. Protein diffusion models operating on such representations were shown to generate designable and novel samples to various degrees (Lin & AlQuraishi, 2023; Ingraham et al., 2022; Watson et al., 2022; Yim et al., 2023). Some of those were additionally designed to condition the sample on properties such as substructure, symmetry or structural motif; however, none of those works link the function to dynamics. Motif scaffolding has been done by, for example, providing the denoised motif residues positions in the conditional training (Watson et al., 2022), by particle filtering methods (Trippe et al., 2023), or by empirically estimating the chances that the sample will have the query motif (Ingraham et al., 2022). Eigenfold (Jing et al., 2023) attempts to incorporate the physical constraints for oscillations into the diffusion kernel, however, it did not improve the sample quality, and it was not tested whether it changes the dynamics of generated samples.

## 2.2 Normal Mode Analysis

Normal Mode Analysis (NMA) is a technique for describing collective motions of protein residues for a given energy function. It assumes that a protein is in the energy minimum state in a given force field, such that the protein residues will, to first approximation, undergo harmonic motions about their minima (Bahar et al., 2010). Amplitudes and frequencies of such oscillations are the solutions to the equations of motions for all residues. These equations of motions are compactly written in matrix form as $\mathbf{M}\ddot{x} = -\mathbf{K}x$, where $x \in \mathbb{R}^{3N}$ is a flattened vector of coordinates of $N$ residues, $\mathbf{M} \in \mathbb{R}^{3N \times 3N}$ is a mass matrix and $\mathbf{K} \in \mathbb{R}^{3N \times 3N}$ is the interaction constants matrix derived from the force field that describes the strength of interactions between residues. Despite the simplistic assumptions about the form of these force fields, NMA has been shown to successfully explain many dynamical phenomena amongst numerous proteins (Gibrat & Gō, 1990; Tama & Sanejouand, 2001; Bahar et al., 1997). Most functional properties of proteins that involve dynamics are related to the low-frequency motions, mathematically represented as the lowest non-trivial eigenvectors of the matrix equation.

## 3 Methods

Consider the following problem: given a target matrix $y_D \in \mathbb{R}^{|\mathcal{C}| \times 3}$, where rows correspond to displacement vectors of $\mathcal{C}$ residues, we aim to generate a new protein in which the displacement vectors of selected residues in their non-trivial lowest normal mode are close to those defined by $y_D$. We use a coarse-grained protein representation, where each residue is represented with the $C_\alpha$ carbon only, and aim to obtain new $C_\alpha$ chains that satisfy the dynamics constraint. To tackle this problem we employ score-based generative modelling (Song et al., 2021). We formulate the agreement of the displacement with a target as a condition in the reverse process and quantify the notion of 'similar dynamics' with a custom loss function.

### 3.1 Conditioning Diffusion Models

The goal of conditional generative modeling is to sample from the posterior $p(x_0|y)$ such that new samples $x_0$ satisfy some chosen property $y$. We specify the following model (Song et al., 2023, Equation 4)

$$p(x_0|y) = \frac{p(x_0)\exp[-l(y, v(x_0))]}{\int p(x_0)\exp[-l(y, v(x_0))]dx_0} \quad \text{and} \quad \kappa(y) = \int p(x_0)\exp[-l(y, v(x_0))]dx_0 \quad (7)$$

where $l(y, v(x_0))$ measures the loss for a measurement of $y$ at $x_0$, $\kappa(y)$ is the normalisation constant, and $v(x)$ maps to the relevant physical quantity represented by $y$. This specification, as shown in Song et al. (2023), allows for guiding a trained unconditional model along the path specified by the loss $l$. Finding an appropriate $p(y|x_0)$ is where the novelty of our method lies. For the dynamics target $y$, if $p(y|x_0)$ was a neural network, it would need to approximate the eigenvectors of an arbitrary symmetric matrix. To the best of our knowledge, finding matrix eigenvectors for *any* variable size symmetric matrix with a neural network is not considered a solved problem yet (there exist neural network approaches to find eigenvectors, but those require retraining for every new matrix (Gemp et al., 2021; Yi et al., 2004), and are not suitable for a large dataset of backbone structures). A method to reconstruct a graph structure from a set of learned eigenvectors via an interactive Laplacian matrix refinement is presented in Martinkus et al. (2022). However, this approach has never been tested for a reverse reconstruction. We escape the need to train a neural network and equate $p(y|x_0)$ to a simple analytical function.

One of the most common mathematical frameworks to obtain a novel sample with any desired property $y$ consists of estimating conditional scores. Different approximations for estimating said score have given rise to a variety of methods such as classifier guidance (Dhariwal & Nichol, 2021), classifier free guidance (Ho & Salimans, 2022), and 'reconstruction guidance' (Ho et al., 2022; Chung et al., 2022a). What all these approaches have in common is that they decompose the conditional score as

$$\nabla_{x_t} \ln p_t(x_t|y) = \nabla_{x_t} \ln p(y|x_t) + \nabla_{x_t} \ln p_t(x_t), \quad (8)$$

where $p(y|x_t)$ is a probability that the sample meets the condition at $t = 0$ given the state $x_t$ at some other time. Following Chung et al. (2022a), we re-express it with the integral

$$p(y|x_t) = \int p(y|x_0)p_0(x_0|x_t)dx_0. \tag{9}$$

The integral is intractable and we cannot evaluate $p_0(x_0|x_t)$ directly. But as in Chung et al. (2022a), we overcome this via the approximation of the denoiser's transition density with a delta function centred at the mean

$$p_0(x_0|x_t) \approx \delta_{\mathbb{E}[x_0|x_t]}(x_0). \tag{10}$$

Such approximations to the posteriors via point masses centred at their means rather than their modes (MAP) are known as Bayes point machines (Herbrich et al., 2001), and have been shown to outperform MAP. Under this approximation, the entire integral simplifies to

$$p(y|x_t) \approx p(y|\mathbb{E}[x_0|x_t]). \tag{11}$$

Via Tweedie's formula (Chung et al., 2022a), the expected output of the model at $t = 0$ is

$$\mathbb{E}[x_0|x_t] = \frac{x_t + (1 - \bar{\alpha}_t)s(x_t, y)}{\sqrt{\bar{\alpha}_t}}. \tag{12}$$

Under our model specification, via Bayes rule

$$p(y|\mathbb{E}[x_0|x_t]) = p(\mathbb{E}[x_0|x_t]|y)p(y)/p(\mathbb{E}[x_0|x_t]), \tag{13}$$

substituting back into the score we obtain

$$\nabla_{x_t} \ln \left( \frac{p(\mathbb{E}[x_0|x_t])\exp[-l(y, v(\mathbb{E}[x_0|x_t]))]}{p(\mathbb{E}[x_0|x_t])\kappa(y)} p(y) \right) = -\nabla_{x_t} l(y, v(\mathbb{E}[x_0|x_t])). \tag{14}$$

Depending on the quantity $y$, different losses must be used in Equation 14. Note that even though the derivations are done in continuous time, the equivalence of the score and the noise still applies, and we can use the discretised sampling scheme as in Equation 3. Now, we explain our choices for dynamics and structure conditioning losses.

## 3.2 DYNAMICS LOSS

The next step is to define the loss function in Equation 14 that enforces the targeted dynamics while being invariant to the protein rotations and translations. Knowing the expected residues' positions at $t = 0$ and the expected components of the normal mode of the conditioned residues given structure $x_t$ at some time $t$, the invariance is preserved if one compares the relative pairwise angles between the displacement vectors and their relative magnitudes. Moreover, this makes the conditioning target independent of the protein length: eigenvectors are normalised, hence the amplitudes of displacements of a subset of residues depend on the protein length. Therefore, we propose to use the following loss in Equation 14, which is a simple combination of amplitude and angle terms between all pairwise residues. For the rest of this work, we refer to it as the **NMA-loss**.

$$l_{\text{NMA}}(y_D, v(x)) = l_{\text{angle}}(y_D, v(x)) + l_{\text{ampl}}(y_D, v(x)), \tag{15}$$

$$l_{\text{angle}} = \sum_{i,j \in \mathcal{C}} |\cos(y_{D,i}, y_{D,j}) - \cos(v(x_t)_i, v(x)_j)|, \tag{16}$$

$$l_{\text{ampl}} = \sum_{i \in \mathcal{C}} \left| \frac{||y_{D,i}||}{||y_D||} - \frac{||v(x)_i||}{||v(x)||} \right|. \tag{17}$$

In this invariant loss, $y_{D,i}$ and $v(x)_i$ are displacement vectors of residue $i \in \mathcal{C}$ in the target $y_D$ and in the displacements matrix $v(x) \in \mathbb{R}^{|\mathcal{C}| \times 3}$ derived from expected positions at $t = 0$. The amplitude terms are normalised such that only their relative sizes matter, consistent with the fact that amplitude information from NMA can only make relative statements about the participation of a given residue in a mode (Bahar et al., 2010). For the combined loss, in the process of minimisation of NMA-loss in the sampling steps, the $l_{\text{ampl}}$ is scaled by 2, such that its contribution is similar in magnitude to $l_{\text{angle}}$. We compute the NMA-loss using a differentiable implementation of the eigenvector calculations assuming the Hinsen force-field ((Hinsen & Kneller, 1999), more details in Appendix B.2).

### 3.3 STRUCTURE LOSS AND JOINT CONDITIONING

The essential part of our work is building a connection between conditioning on dynamics and conditioning on structure. Even though dynamics and structure are correlated, many structures will have similar low-frequency eigenvectors, and there is no guarantee that the particular protein packing will correspond to the biological function for which the dynamics were designed. Therefore, dynamics conditioning must be accompanied by structure conditioning. Structure conditioning enforces the generated protein backbone to have a subset of residues $C_M$ positioned in pre-defined relative positions. For example, structure conditioning might enforce the presence of a given functional motif $M$ somewhere in the arbitrarily rotated protein. We denote the target positions as $y_M \in \mathbb{R}^{|C_M| \times 3}$, and $x_{C_M} \in \mathbb{R}^{|C_M| \times 3}$ is the prediction of conditioned residues' positions at $t = 0$ in the sampling process. In the language of score-based generative modeling, the conditional score for the joint target $(y_D, y_M)$ will be now decomposed into three terms

$$\nabla_{x_t} \ln p_t(x_t | y_D, y_M) = \nabla_{x_t} \ln p(y_D | x_t) + \nabla_{x_t} \ln p(y_M | x_t) + \nabla_{x_t} \ln p_t(x_t). \quad (18)$$

Finally, the appropriate **structure loss** should be substituted to the $\nabla_{x_t} \ln p(y_M | x_t)$ term. We define the structure loss to be the misalignment between $y_M$ and $x_{C_M}$, specifically the $L1$ loss between all $C_M$ residues' coordinates. In order not to violate equivariance, we use our custom differentiable implementation of the Kabsch algorithm (Kabsch, 1976; 1978) to find the best fit of the target residues $y_M$ and $x_{C_M}$ at the reverse diffusion step and only then compute the misalignment. In the discussion of the results, we report the final root-mean-square deviation (RMSD), which is related to but different from the structure loss (see Section 5.2).

## 4 MODELS AND THE EXPERIMENTAL SETUP

The aim of the experimental evaluation is two-fold. Firstly, we test whether the proposed conditioning method indeed results in better agreement between the target and the novel structure's dynamics. To do so, we use our custom denoiser model, perform conditional sampling using a large number of dynamics targets and examine the conditioning effectiveness. Secondly, we utilise Genie (Lin & AlQuraishi, 2023), the diffusion model able to produce high-quality samples and modify its sampling scheme with our joint conditioning. We therefore demonstrate the universality of our framework which leaves an open path to transferring our method to other large protein diffusion models. The modified Genie model produces samples conditioned on the hinge targets which we thoroughly evaluate for designability.

### 4.1 MODELS

**GVP** (Geometric Vector Perceptron (Jing et al., 2021b;a)) is the main building block of our equivariant denoiser. We use a Graph Neural Network with 5 layers based on GVP (details in the Appendix B.1). The denoiser was trained with the loss function given by Equation 4. We use the Hoogeboom schedule (Hoogeboom et al., 2022) with a 250-step DDPM discretisation scheme. The model was trained for 1000 epochs with a learning rate of 1e-4.
**Genie.** Genie (Lin & AlQuraishi, 2023) is a diffusion probabilistic model with the DDPM discretisation. It takes advantage of the protein geometry by extracting the Frenet-Serret frames of residues at each noise prediction step, which are then passed to the SE(3)-equivariant denoiser. Genie outperformed other models such as ProtDiff (Trippe et al., 2023), FoldingDiff (Wu et al., 2022) or FrameDiff (Yim et al., 2023), and remains comparable to RFDiffusion (Watson et al., 2022). For our experiments, we used the published weights of the model trained on the SCOPe dataset (Fox et al., 2014; Chandonia et al., 2021) able to work with proteins up to 256 residues long.

### 4.2 DATASET AND TARGETS

For our custom model training, we extract all short monomeric CATHv4.3 domains (Orengo et al., 1997) for structures with high resolution ($< 3$Å), of lengths between 21-112 amino acids, clustered 95% sequence similarity to remove redundancy. The resulting dataset contained 10037 protein structures.
We extract *random* and *strain* dynamics targets from the proteins in the validation set. Random targets are the displacements in the randomly chosen sets of 10 consecutive residues; for the strain

targets, we perform strain-energy calculation (Hinsen & Kneller, 1999) (details in the Appendix B.2) and choose 10 consecutive residues with the largest summed energy.

Joint conditioning imposes constraints on both the protein normal mode and the specific residues' positions. Biologically relevant targets that require such constraints are the hinge parts of proteins. Three proteins were selected from the literature: lysozyme (PDB ID: 6lyz), adenylate kinase (PDB ID: 3adk), and haemoglobin (PDB ID: 2hhb). In each protein we analysed which residues participate in the hinge motion – those residues constitute the $y_M$ targets. For each protein we perform NMA calculation to obtain the displacements of the hinge residues – the $y_D$ targets (details in Appendix D).

## 4.3 EVALUATION METRICS

**Population level.** For the first set of experiments investigating dynamics conditioning, we focus on quick-to-compute statistics of the large sample set to understand the expected effects of conditioning on the sample quality. Apart from the NMA-loss, we check the sample quality using: (1) the mean chain distance ($C_\alpha - C_\alpha$) that should be close to 3.8 Å (2) the radius of gyration of the backbone, which is an indicator of whether the model produces samples with an adequate compactness; (3) secondary structure statistics (SSE), that is, the proportion of $\alpha$-helices, $\beta$-sheets and disordered loops; (4) novelty in terms of the TM-score to the closest structure in the train set. TM-score measures the topological similarity of protein structures and has values in the range $[0, 1]$. TM-score $> 0.5$ suggests two structures are in the same fold (Xu & Zhang, 2010).

**Detailed statistics.** In the case of joint conditioning, we sample novel protein backbones using Genie and check the designability of the new samples using the same *in silico* evaluation pipeline as in benchmarking unconditional Genie. For each backbone sample, we obtain 8 ProteinMPNN generated sequences and fold each sequence with ESMFold (Lin et al., 2022). We calculate the self-consistency TM-scores (scTM), that is, the TM-scores between the input structure and each of the ESMFold predictions. scTM scores were also considered in other works (Trippe et al., 2023; Lin & AlQuraishi, 2023) as one of the standard metrics for sample quality evaluation. We report the proportion of conditional samples whose best scTM-score to one of the ESMFold designed structures is $> 0.5$, in the same fashion as in Trippe et al. (2023) that tackles a similar motif conditioning problem.

## 4.4 SAMPLING DETAILS.

**Dynamics conditioning with GVP.** The sampling process consisted of 250 reverse diffusion steps (details in the Appendix B.3). We extracted 300 strain and 300 random targets from 300 randomly sampled proteins from the validation set. For each target, we took 3 conditional and unconditional samples, and for each group we selected the one with the lowest NMA-loss. Each sample had the same length as the protein from which the target was extracted.

**Joint conditioning with Genie.** The original Genie sampling loop with 1000 time steps in the generation was modified to include the conditional score (details in the Appendix B.3). The guidance scales were different for each target, and in the order of 2000-3000.

## 5 RESULTS AND DISCUSSION

### 5.1 STRAIN AND RANDOM DYNAMICS TARGETS

Here we present the results for the strain and random dynamics targets. At the start, we filter out the 'low quality' samples that evidently do not form a biologically valid proteins (details in the Appendix C.We examine if the conditioning has the desired effect of enforcing the target normal mode. Figure 2 shows that indeed, the NMA-loss is successfully minimised in the conditional samples as compared to the unconditional ones. Note that both the target normal mode and the mode of the newly sampled structure must obey some physical constraints imposed on all proteins and the degrees of freedom of all relative displacements are limited, therefore it is occasionally possible to obtain low loss for the unconditional sample. Encouraged by this finding, we proceed to the visual inspection of the samples. Figure 3 shows a pair of conditional and unconditional samples for one of the strain targets (additional sampled pairs are in Appendix G). There is a better

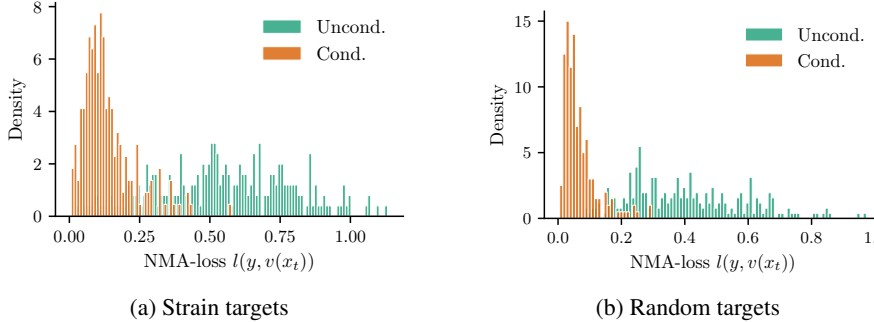

| (a) Strain targets | (b) Random targets |

Figure 2: Density histograms of the NMA-loss for the dynamics conditioning using random and strain targets. Conditioning shifts the distribution towards lower values, such that the distribution has an evident sharp peak.

alignment of the displacement vectors and target vectors for the conditional sample as compared to the unconditional one, which we also consistently observed for the rest of the sampled pairs. We conclude that our conditioning has the desired effect of enforcing the target dynamics. We therefore proceed to the quality check of the samples – we must ensure the conditioning does not compromise the backbone structure. To ensure that the sampled proteins are still biologically valid, we evaluate

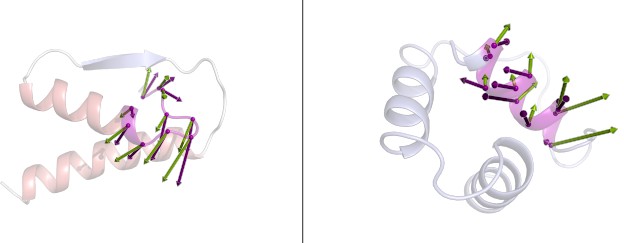

Figure 3: Comparison of two samples for the same strain target. **Left :** Conditional sample (NMA loss-0.114). **Right :** Unconditional sample (NMA-loss 0.740). In the conditional sample, green vectors (new displacements) have much more similar relative amplitudes and pair-wise angles to the purple vectors (target) when compared to the unconditional sample. In both left and right visualisation, purples were rotated to match greens.

their geometry. In the end, we investigate the samples' novelty to check whether the diffusion model has not simply memorised the train set.

Figure 4 shows the SSE and $R_g$ of the samples compared to the train CATH dataset. Unconditional samples show a variety of SSE in proportions close to the CATH dataset. Interestingly, we found that conditioning increases the proportion of $\beta$-sheets at the expense of $\alpha$-helices. $R_g$ distributions of both unconditional and conditional samples have a visible overlap with the CATH $R_g$ distribution, the second one is shifted to larger values (but remains within the $R_g$ values observed in CATH). Therefore, while the conditional samples do not violate physical constraints, the dynamics conditioning introduces changes in protein packing. Whether this effect is significant for downstream applications when the conditioning is transferred into problem-specific models is left for future work. Respective Figures for the random targets can be found in Appendix A. Lastly, we calculate the novelty of the samples expressed in terms of TM-score to the closest structure in the train set. Both unconditional and conditional samples of both target types were highly novel, with TM-score lower than 0.5 in 90% of the samples.

## 5.2 HINGE TARGET

Finally, we present results for the joint conditioning. The conditional samples were filtered using criteria of mean chain distances outside [3.75, 3.85] Å interval and RMSD with respect to the motif smaller than 1 Å. These constraints left us with 43%, 60% and 23% of the conditional samples for

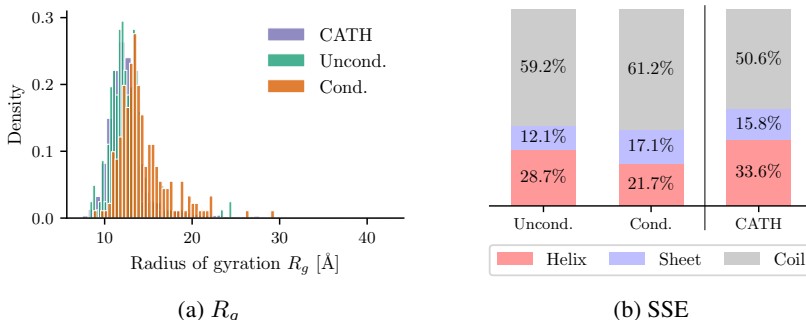

| (a) $R_g$ | (b) SSE |

Figure 4: Density histogram of $R_g$ and SSE proportions for strain targets.

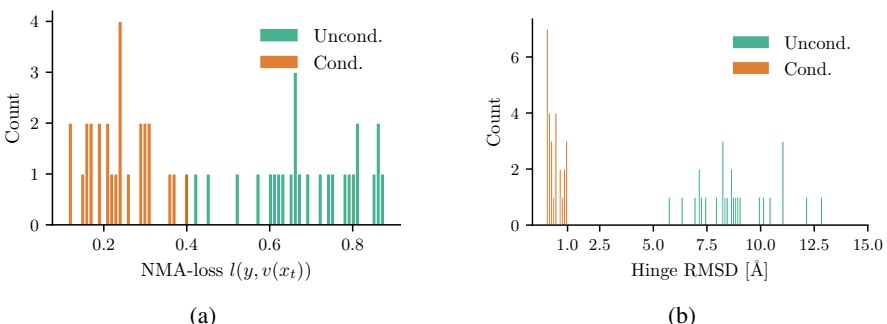

| (a) | (b) |

Figure 5: NMA-loss and RMSD for the lysozyme hinge target. Conditional samples achieve low values of NMA-loss and RMSD that none of the unconditional samples have.

lysozyme, adenylate kinase and haemoglobin, respectively, such that we ended up with 27 conditional samples. To match that number, we sampled 27 unconditional ones. In the analysis of the remaining samples, we considered the distributions of NMA-loss (see Figure 5) and scTM-score. The distribution of the NMA-loss confirms that our method can enforce the specific dynamics and conditions on the structure at the same time. Analysis of the designability revealed that the distribution of scTM-scores depends on the target we use. The proportions of conditional samples with scTM-score > 0.5 were 0.48, 0.78, 0.41 for lysozyme, adenylate kinase and haemoglobin, respectively. Interestingly, when we sampled 27 structures just with the hinge dynamics conditioning, those values were 0.93, 1.0, and 0.89, respectively, and the decrease in designability can be attributed purely to the difficulties in the structure conditioning (Appendix E). Additional experiments with a conditionally trained Genie model and extra designability results can be found in Appendix H. We finish with the visual investigation of the generated hinge structures. Figure 1 shows pairs of the targets and the new samples (more examples in the Appendix G). The new samples indeed possess the hinge structure, as well as the hinge-like low-frequency motion.

## 6 CONCLUSIONS AND FURTHER WORK

For the first time, we condition the protein diffusion model on dynamics, thus paving the way to designing more functional proteins in the future. We also make the code publicly available[1]. We generate novel proteins with a pre-defined lowest non-trivial normal mode of oscillation for a subset of residues. The large-scale statistics show that the conditioning is effective and can be transferred to already trained unconditional models. The extended version of the conditioning that includes the structure conditioning is implemented as part of the unconditional Genie model and we produce novel proteins that exhibit hinge structure and dynamics while remaining designable by the scTM

---

[1]Code available at https://github.com/ujk21/dyn-informed.

criteria. Further work includes integrating the dynamics conditioning with other types of structure conditioning, and further evaluation with other types of motions.

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

## APPENDICES

## A    POPULATION STATISTICS FOR RANDOM TARGETS

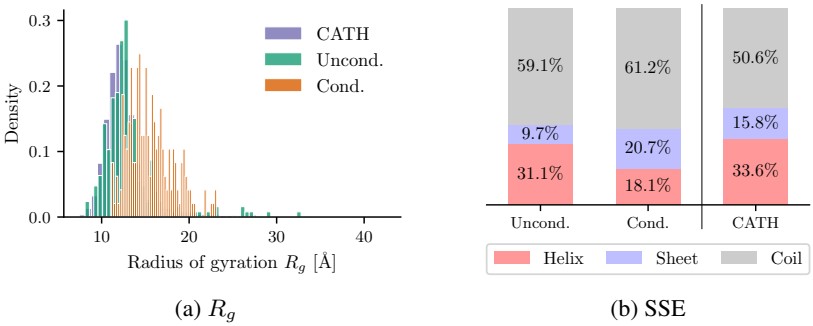

(a) $R_g$   (b) SSE

Figure 6: Density histogram of $R_g$ and SSE proportions for random targets.

# B IMPLEMENTATION DETAILS

## B.1 CUSTOM MODEL DETAILS

The custom model was based on the Geometric Vector Perceptron-Graph Neural Network architecture Jing et al. (2021b;a). Each protein was represented as a fully connected graph. The node scalar features were sinusoidal positional embeddings of the residues' order in the chain concatenated with a normalised time step feature. We perform a message-passing on a fully connected graph of $C_\alpha$ carbons. Edge features were distances between nodes in terms of 16 Gaussian radial basis functions and the unit vectors pointing along the edge. The model used 5 GVP-Convolutions layers and the output of the network (the noise) had the centre of mass subtracted to ensure equivariance.

## B.2 NMA CALCULATIONS DETAILS

Nowadays quick and ready-to-use implementations of NMA are available, such as the Biotite extension Springcraft (Kunzmann & Hamacher, 2018), which we used and rewrote into a PyTorch differentiable version.

Before any equations of motion can be written, one must specify a force field that describes interactions between residues. We use a Hinsen force-field (Hinsen & Kneller, 1999) with a cutoff of 16 Å. For the choice of the strain target we perform the strain-energy calculation as described in Hinsen & Kneller (1999)

$$E_i = \frac{1}{2} \sum_j^N k(\mathbf{R}_{ij}) \frac{|(\mathbf{d}_i - \mathbf{d}_j) \cdot \mathbf{R}_{ij}|^2}{|\mathbf{R}_{ij}|_2^2} \tag{19}$$

where $E_i$ is the energy of residue $i$, $\mathbf{R}_{ij}$ is a vector that is the equilibrium separation between the residues $i, j$, $k(\mathbf{R}_{ij})$ is the interaction constant, and $\mathbf{d}_i, \mathbf{d}_j$ are the displacements of residues $i, j$ in the mode to be analyzed (here, the lowest non-trivial normal mode).

## B.3 SAMPLING DETAILS

### B.3.1 GVP

The generation was run with 250 reverse time steps, but at the last two generation time steps the noise in the update step was set to 0, since we found that this results in chain distances remaining closer to 3.8 Å. It is a common practice to upscale the conditional term $\nabla_{x_t} \ln p(y|x)$ by some guidance scale (Dhariwal & Nichol, 2021). Guidance scales for strain targets were time-dependent and equal to $200\alpha_t$ for strain targets and $400\alpha_t$ for random targets. Conditioning was switched on in the middle of the generation process. Since each sample had the same length as the protein from which the random or strain target was extracted, the potential differences observed in SSE cannot be attributed to differences in the protein length distributions.

### B.3.2 GENIE

To take samples with the Genie model we used an additional parameter $\eta$ to downscale the noise in the reverse process, as recommended in the Genie publication (Lin & AlQuraishi, 2023). We set $\eta = 0.4$ which was shown to achieve the best trade-off between designability and diversity. We found achieving the balance between the conditional parts of the score for dynamics and for the structure to be the most problematic aspect to optimise. With the means of trial and error fine-tuning of the guidance scales, we arrived at different values per each hinge target. The guidance scales for the dynamics term and structure term were 3000 and 2500 for 6lys; 3000 and 2000 for 3adk; 2500 and 2000 for 2hhb. These constants were scaled by the time-dependent factors: $\alpha_t$ for dynamics and $\sqrt{1.5 - \alpha_t}$ for structure. Since we fine-tuned the guidance scales and their time dependencies, we skipped the $\sqrt{1 - \bar{\alpha}_t}$ factor when converting score to noise. The conditioning was switched on in the middle of the generation. Within the Kabsch algorithm in the structure conditioning, we found the translation vector and rotation matrix to get the best alignment of the target residues with the residues' positions at $t = 0$, applied those transformations to the target and calculated RMSD. The translation and rotation were recalculated each 5 time steps.

## C    LOW QUALITY SAMPLES

The 'low quality' samples are those where the mean chain distance is outside [3.75, 3.85] Å interval (proteins with mean chain distance more extreme being rare in nature (Voet & Voet, 2010)). Occasionally during conditional sampling coordinate values increases by orders of magnitude along the sampling trajectory, or even explodes to NaN values. This divergence effect has also been observed in many conditional diffusion models (Lou & Ermon, 2023); in our case, this tends to happen when the conditioning pushes a sample's coordinates outside of the realm of observed samples for which the denoiser was trained. Finding the right balance between the NMA-loss driven part of the score and the unconditional part of the score is an important part of the conditioning process. These diverged samples are also filtered out during the evaluation process. In the end, about $20\%$ of samples in each category were filtered out due to low-quality chain distances.

## D    HINGE TARGETS DESCRIPTION

To extract targets with a prominent hinge motion we performed a literature survey. We identified the lysozyme (Gibrat & Gō, 1990), adenylate kinase (AdK) (Tama & Sanejouand, 2001), and haemoglobin (Perahia & Mouawad, 1995) as three prominent examples of proteins with hinge-type motions for which the lowest normal mode is also known to correlate strongly with functional motion. To extract the hinge motion, we perform an anisotropic elastic network formulation of normal mode analysis with an invariant force field on alpha carbon atoms, using a distance cut-off of 13 Å. The lowest non-trivial normal mode is then computed from the Hessian, and the 16 residues with the largest displacement components are extracted as motifs to scaffold with the targeted motion. The target motifs are shown in the top column of Fig. 1. For lysozyme and adenylate kinase hinges, the newly sampled backbones had length $\max(\text{hinge\_residue\_order}) + 10$, and for the haemoglobin $\max(\text{hinge\_residue\_order}) + 20$, where (hinge\_residue\_order) is the order of non-consecutive hinge residues in the original backbone. Since haemoglobin is larger than the maximal backbone length that can fit to genie, the haemoglobin hinge was modified - the backbone order for all hinge residues was shifted down by 190 residues, additionally the number of residues between the hinge arms was decreased by 250 residues.

## E    DESIGNABILITY IN DYNAMICS CONDITIONING VS STRUCTURE CONDITIONING

Since experimental verification of protein designs is time-consuming and expensive, the research community has developed *in silico* methods to assess design success computationally. Many of them fall under the framework of so-called *self-consistency metrics* (Trippe et al., 2023), meaning that the designed structure is evaluated by predicting a sequence for it via inverse folding models like ProteinMPNN (Dauparas et al., 2022), predicting the resulting structure via structure prediction methods like AlphaFold2 (Jumper et al., 2021) or ESMFold (Lin et al., 2022) and comparing this predicted structure to the designed one via structural similarity metrics.
The most common computational design criteria are the following:

- scTM $> 0.5$: the TM-score between the designed structure and the self-consistency predicted structure as described above. With the scTM-score ranging from 0 to 1, higher numbers correspond to an increased likelihood of the input structure being designable. A threshold of 0.5 is often chosen and the percentage of samples above this threshold is reported.
- scRMSD $< 2$ Å: The scRMSD metric is similar to the scTM metric, however instead of the TM-score the RMSD between the designed and predicted structure is calculated. It is a much more stringent criterion than scTM since RMSD is a local metric that is more sensitive to small structural differences.
- pLDDT $> 70$ and pAE $< 10$: Since both scTM and scRMSD rely on a structure prediction method like AlphaFold2 to be reliable metrics, confidence metrics of these models like pLDDT and pAE are used as additional metrics to ensure the reliability of self-consistency metrics. Low scRMSD and high pLDDT have been linked to the experiment success of designing the backbone (Bennett et al., 2022).

scTm score alone is a good indicator of whether two structures are in the same fold and for that reason, it has been used in previous works for assessing the general sample quality (Trippe et al., 2023; Yim et al., 2023). However, more recent works such as Genie (Lin & AlQuraishi, 2023) apply more stringent criteria of pLDDT > 70, pAE < 10, scTM > 0.5. In our experiments with hinge targets, if those additional requirements were incorporated, the proportion of samples meeting those criteria in joint conditioning dropped to 0.04 for 6lys, 0.15 for 3adk, and 0.0 for 2hhb. When we incorporated the last most stringent criterion that scRMSD < 2 Å, those proportions dropped to 0.00, 0.04 and 0.0. Further investigation revealed that the lack of confidence in ESMFold predictions is due to the difficulty in structure conditioning. When only the dynamics conditioning was used (with the same guidance scale as when being part of the joint conditioning) the proportions of designable structures without scRMSD criterion were 0.6 for 6lys, 0.82 for 3adk, 0.52 for 2hhb, and with scRMSD criterion 0.41, 0.63 and 0.37 respectively.

To put these values into perspective, we note that low designability scores are not uncommon for the models tackling motif scaffolding problem. The current state-of-the-art model, RFDiffusion, has designability 0, or close to 0, for some of the more difficult functional site targets ((Watson et al., 2022), Supplementary Methods Table 10). Since our targets were extracted from a flexible part of the protein, consist of discontinuous motifs and have not been used as targets in the literature elsewhere, it is difficult to assess what designability scores might be considered 'good' for those targets. Moreover, we note that the confidence metric of AF2/ESMFold might not be well suited for the assessment of the quality of the flexible regions. As observed in Bryant (2023), pLDDT is a 'good' metric if a single protein conformation is considered, however, it becomes less informative as alternative conformations are included. The regions with lower pLDDT tend to be flexible regions with conformational changes, which might explain why proteins with a hinge structure tend to have lower pLDDT.

## F    SCORE-NOISE EQUIVALENCE

For completeness, we provide a short derivation of the score-noise equivalence

$$\nabla_{x_t} \log q(x_t|x_0) = \nabla_{x_t} \log \mathcal{N}\left(x_t; \sqrt{\bar{\alpha}_t}x_0, (1-\bar{\alpha}_t)I\right) \tag{20}$$

$$\nabla_{x_t} \log \mathcal{N}\left(x_t; \sqrt{\bar{\alpha}_t}x_0, (1-\bar{\alpha}_t)I\right) = -\nabla_{x_t}\frac{(x_t - \sqrt{\bar{\alpha}_t}x_0)^2}{2(1-\bar{\alpha}_t)} \tag{21}$$

$$-\nabla_{x_t}\frac{(x_t - \sqrt{\bar{\alpha}_t}x_0)^2}{2(1-\bar{\alpha}_t)} = -\frac{(x_t - \sqrt{\bar{\alpha}_t}x_0)}{(1-\bar{\alpha}_t)} \tag{22}$$

$$-\frac{(x_t - \sqrt{\bar{\alpha}_t}x_0)}{(1-\bar{\alpha}_t)} = -\frac{\epsilon_t}{\sqrt{1-\bar{\alpha}_t}} \tag{23}$$

## G    ADDITIONAL SAMPLES

Green arrows are the displacements of conditioned residues in the sampled protein, purple arrows are the targets rotated to fit the green arrows best.

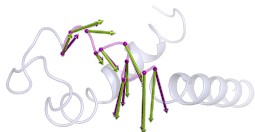 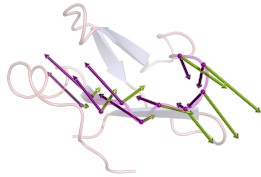

(a) Conditional sample (NMA-loss 0.088)    (b) Unconditional sample (NMA-loss 0.620)

Figure 7: Comparison of the conditional and unconditional sample for the same strain target.

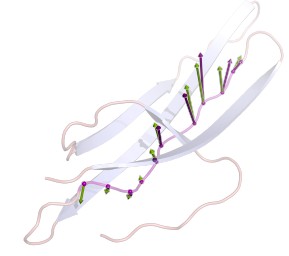 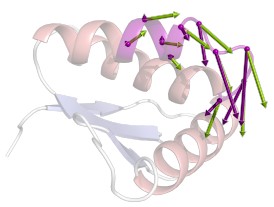

(a) Conditional sample (NMA-loss 0.224)    (b) Unconditional sample (NMA-loss 0.625)

Figure 8: Comparison of the conditional and unconditional sample for the same **random** target.

## H  JOINT STRUCTURAL MOTIF & DYNAMICS CONDITIONING WITH THE IMPROVED GENIE MODEL

When using the original, guidance-based formulation for motif conditioning used in the main text, we found that motif conditioning continued to be the primary difficulty. This made it harder to perform and analyse NMA conditioning jointly with motif conditioning, because the NMA condition only makes sense for a reasonably well formed motif.

We therefore sought to improve motif conditioning by using a Genie model that we re-trained with the explicit motif conditioning, as proposed in Didi et al. (2023).

### H.1  TARGET DEFINITION

We investigate the following question: Can we design a new backbone, such that the functional motif **and** its key dynamical behavior, represented by the lowest non-trivial normal mode components of the motif in the target structure, are preserved?

To test out a biologically relevant scenario we choose to model a dynamically relevant segment spanning the two active site residues in hen-egg white lysozyme as target motif. Lysozyme was chosen as a case study since during function (Bauer et al. (2019); Brooks & Karplus (1985)) it undergoes a well-studied hinge motion, which is well captured by lowest non-trivial mode in normal mode analysis. The motif is illustrated in Fig. 10 and consists of 22 residues of the original structure (PDB: 6lyz, 129 residues), including the active site residues GLU-35 and ASP-52 (Vocadlo et al. (2001); Held & van Smaalen (2014)). To obtain the NMA target for this motif, we perform an NMA with an invariant force-field and with a 13 Å distance threshold on the $C_\alpha$-backbone of the native protein (6lyz) and extract the lowest non-trivial normal mode displacements for the motif residues as NMA target.

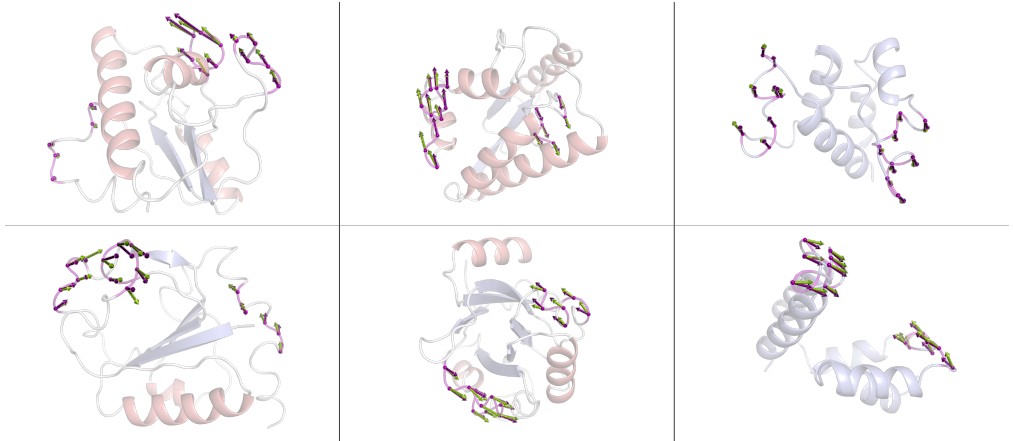

Figure 9: More samples with joint conditioning. **Left** column: 2 samples for the 6lys target. **Middle** column: for 3adk target. **Right** column: for the 2hhb target.

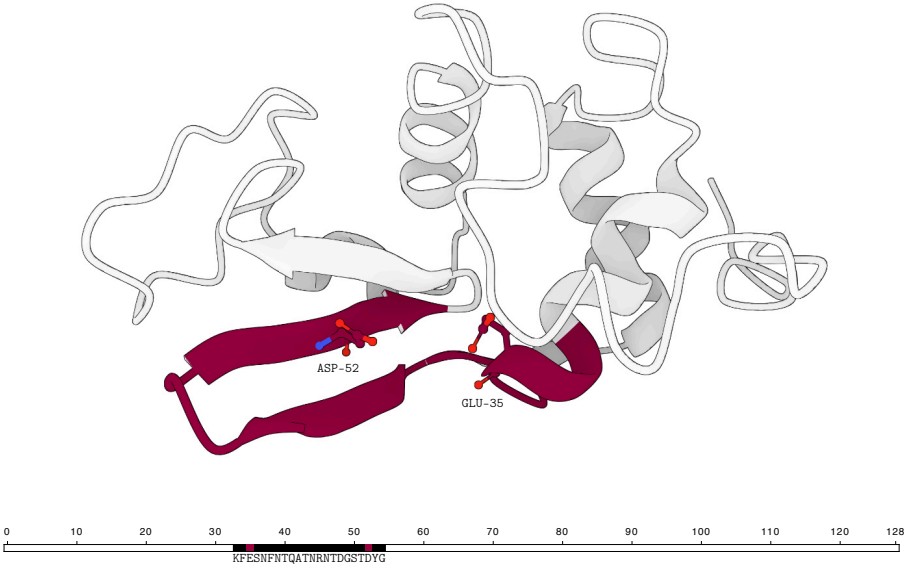

Figure 10: Target definition for the additional experiments and to illustrate a biological application. The target motif (red) was chosen as the single segment connecting the two active site residues (GLU-35, ASP-52) of hen-egg white lysozyme (PDB:6lyz), including two residues on either side of the active site. This results in a target motif of 22 residues in length. The active site residues are shown with side-chains, and the motif's position in the overall sequence is marked on the bottom bar.

## H.2 MODELLING

**Improved motif conditioning model** We modify the unconditional Genie model (Lin & AlQuraishi, 2023) in order to perform the conditional training, where the model is provided with the motif coordinates for some of the training examples. We add an additional conditional pair feature network that takes the target motif coordinates and frames as input with zero-padding for all non-motif coordinates and frames. The features of this motif-conditional pair feature network are fused with the output of the original unconditional pair feature network in genie via concatenation along the feature dimension, followed by a linear projection down to the channel size of the unconditional model. The remainder of the Genie model then proceeds unchanged. This minor architectural modification means our conditional Genie network has $4.162$M parameters while the unconditional

Genie network has 4.087M parameters ($\sim 1.8\%$ fewer). The conditional Genie model was trained for 4'000 epochs on 4 A100 GPUs ($\sim 300$ A100 hours in total). We stopped training at this point, as we observed almost comparable performance to the publicly available model weights (which were obtained after training for 50'000 epochs). We use these model for all the additional experiments in this section. The model is trained according to algorithm 5 in Didi et al. (2023) and during training the model is shown a conditional sample for $80\%$ of the time and an unconditional one for the remaining $20\%$.

**Guidance schedule** While motif scaffolding is now explicitly built into the denoiser, we still need to condition on the NMA dynamics condition. We follow a reconstruction guidance (Chung et al., 2022b) approach with a modulated step-function guidance schedule

$$\gamma(t) = \begin{cases} \gamma_0(1 - \alpha_t) & \text{if } t < t_{\text{start}} \\ 0 & \text{if } t \geq t_{\text{start}} \end{cases}, \tag{24}$$

with a guidance scale $\gamma_0$ and starting point $t_{\text{start}}$. For $t_{max} = 1000$, we fixed $t_{\text{start}} = 500$ (i.e. conditioning starts halfway through the reverse diffusion process) and identified $\gamma_0 = 500$ as an adequate guidance scale through a logarithmic scan of $\gamma_0$ values. Similar to other work on diffusion models for protein backbone generation, we reduce the noise scale by a factor $\eta = 0.4$, which improves the quality of generated samples (Yim et al., 2023) for motif-only as well as motif+NMA conditioning.

**NMA loss** The presence of a functional motif defines a reference coordinate system, namely the coordinate system in which the coordinates of the to-be-scaffolded motif are given in. Notably, this means that the normal mode displacements at the motif residues are also given in the motif's coordinate system. Any designed backbone should be *invariant* to translations, but *equivariant* to rotations of this coordinate system, which correspond to rotations of the motif and the associated displacement vectors.

To better comply with these symmetry requirements, we adapt the invariant loss $l_{\text{NMA}}$ in Eq. 15 to make use of this reference coordinate system. Using the notation of the main text, $y^M \in \mathbb{R}^{m \times 3}$ and $x^M \in \mathbb{R}^{m \times 3}$ represent the target motif coordinates and sample motif coordinates for a motif of $m$ residues respectively. Similarly, $v^M(y) \in \mathbb{R}^{m \times 3}$ and $v^M(x) \in \mathbb{R}^{m \times 3}$ respectively refer to the matrix of displacement vectors in the lowest non-trivial normal mode for the target and the sample. The rotation matrix $\mathbf{R}(y^M, x^M)$ transforms the coordinate frame of the target motif $y^M$ to that of $x^M$. With these definitions, the updated NMA loss for the additional experiments is

$$l'_{\text{NMA}} = 2l_{\text{direction}}\left(\mathbf{R}(y^M, x^M)v^M(y), \ v^M(x)\right) + l_{\text{magnitude}}\left(\mathbf{R}(y^M, x^M)v^M(y), \ v^M(x)\right) \tag{25}$$

$$l_{\text{direction}}(v_1, v_2) = 1 - \left| \frac{v_1}{\|v_1\|} \cdot \frac{v_2}{\|v_2\|} \right| = 1 - |\cos(v_1, v_2)| \tag{26}$$

$$l_{\text{magnitude}}(v_1, v_2) = |\|v_1\| - \|v_2\||. \tag{27}$$

Here, $v(y)$ and $v(x)$ are understood as flattened vectors in $\mathbb{R}^{3m}$, and therefore $l_{\text{direction}}$ directly captures the relative contributions of each residue's displacement. The factor 2 was added to align the min and max ranges of the two components of $l'_{\text{NMA}}$. We obtain $\mathbf{R}(y^M, x^M)$ through a differentiable implementation of the Kabsch alignment algorithm (Kabsch, 1976) and $v(x)$ from a differentiable implementation of NMA on the sampled backbone $x$, which is then subset to the motif coordinates, as in the main text. Guidance is then performed via

$$x_t \leftarrow x_t - \gamma(t)\nabla_{x_t}l'_{\text{NMA}}\left(\mathbf{R}(y^M, \hat{x}_0^M(x_t))v^M(y), \ v^M(\hat{x}_0(x_t))\right), \tag{28}$$

with $\hat{x}_0(x_t)$ indicating the current estimate of the denoised structure via Tweedie's formula (Robbins, 1956) as in reconstruction guidance (Chung et al., 2022b).

**Evaluation pipeline** The evaluation proceeds similarly as in the main body. For each $C_\alpha$-only backbone sample, we sampled 8 sequences with ProteinMPNN ($T_{\text{sampling}} = 0.1$). In those sequences, the amino acid identities of the motif residues known from the lysozyme target were kept fixed, such that only the scaffold was predicted by ProteinMPNN. Each of the 8 sequences is then re-folded with ESMFold, and self-consistency scores (scNMA, scRMSD, scTM) are calculated with respect to the original backbone sample. The original backbone sample is then paired with the ESMFold-ed design

that had the lowest scRMSD (out of 8 ESMFold designs). We deemed the structure designable if it met the criteria of scTM>0.5, scRMSD<2Å, with confidence threshold of pLDDT>70, pAE<10 for ESMFold predictions, which aligns with definitions in prior work (Watson et al., 2022; Yim et al., 2023; Lin & AlQuraishi, 2023). Moreover, we evaluate an additional motif scaffolding metric, scMOTIF-RMSD, which measures the RMSD between the motif residues in the designed structure (after sequence design & ESMFold) and the target motif.

## H.3 RESULTS

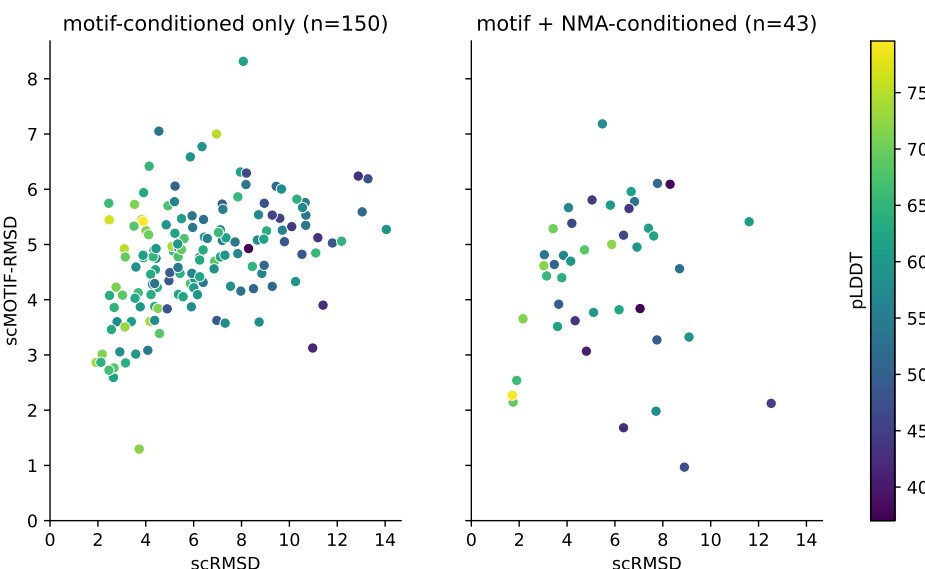

Figure 11: scMOTIF-RMSD vs scRMSD for 150 motif-only and 43 motif+NMA samples (not filtered for designability criteria). While the chosen target (two paired anti-parallel beta sheets connected to the end of a helix) turns out to be a difficult problem for the model, the best samples that achieve lowest scMOTIF-RMSD and scRMSD values stem from joint motif+NMA conditioning, highlighting that the dynamics-conditioning can successfully work the motif scaffolding.

The analysis of the NMA loss of the Genie generated $C_\alpha$-only backbone and scNMA-score of the ESMFold design confirmed that the dynamics conditioning indeed results in the $C_\alpha$ backbones that match the target, however there is no clear direct correspondence scNMA-scores to the original backbone NMA-loss for the dynamics-conditioned samples. Surprisingly, we found while motif conditioning improved upon making the structure conditioning inherent to the conditional Genie model, performing the motif scaffolding was still challenging to the model. In the remainder of this discussion, we call all the structure-only conditioned samples **motif-only**, and all jointly structure and dynamics conditioned - **motif+NMA**.

**Discussion of the motif scaffolding success rate**   As a first part of the evaluation, we calculated the proportion of Genie generated backbones of the motif-only and motif+NMA samples that meet the designability criteria and have backbone design motif-RMSD< 1Å. Out of 150 motif-only samples, 1 is designable and has motif-RMSD< 1Å, while 2 out of 43 motif+NMA ones are. Moreover, the scMOTIF-RMSD, that is RMSD to the motif structure after folding inferred sequences for said backbone with ESMFold, does not achieve values lower than 1Å for any of motif+NMA and motif-only conditioned samples. Figure 11 shows in detail how scMOTIF-RMSD correlates with scRMSD.

We believe this is a combination of (1) the limited training and capacity of our model and (2) the challenging nature of our target motif, which is a segment of 2 paired, anti-parallel beta-sheets connected to the end of a helix. To these points into context, our model was trained for the motif-scaffolding task for 300 A100 GPU-hours, compared to state-of-the-art models such as RFDiffusion, which are trained for over 25′000s of GPU hours when considering the RosettaFold2 pre-training.

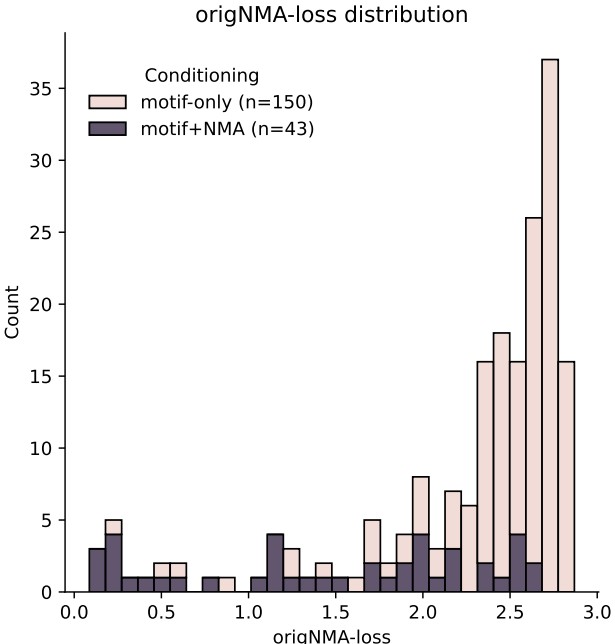

Figure 12: Distribution of the NMA-loss of the designed backbones. The distribution of the motif+NMA conditioned backbones is strongly enriched towards low NMA-loss values when compared to the distribution of motif-only backbone samples. Only 2/150 ($\sim 1.3\%$) of motif-only backbones achieve origNMA-loss $< 0.5$, as opposed to 10/43 ($\sim 23\%$) for motif+NMA – corresponding to a roughly 17x-fold enrichment, while achieving comparable motif scaffolding performance (c.f. Fig. 11). Note that the bars were stacked to avoid overlapping bars from being invisible.

Yet, despite the significantly higher model capacity of RFDiffusion (42 Mio. parameters) as well as the longer training, design success rates (according to the criteria outlined above) of RFDiffusion can also at or below 1% for some challenging, contiguous functional motifs (e.g. targets 5WN9 or 4JHW in the RFDiffusion benchmark in the supplementary material of Watson et al. (2022)). It is therefore possible that the in-silico success rates for our model with a lower capacity are be below the detection threshold for this particular motif scaffolding problem.

Nonetheless, the ESMFold designed backbones achieving the lowest scMOTIF-RMSD and the lowest scRMSD belong to the motif+NMA conditioned group, which illustrates that our NMA-conditioning approach has no discernable negative impact on the designability of samples. We believe it is therefore still meaningful to gleam insights from this set of samples, despite the challenging nature of the motif-scaffolding for our chosen target.

**Discussion of the scNMA-score**   The distribution of NMA-loss in the motif+NMA and motif-only Genie backbone samples is consistent with our previous findings that the dynamics-conditioning leads to the targeted dynamics in the raw backbone (Figure 12). Only 2/150 ($\sim 1.3\%$) of motif-only backbones achieve origNMA-loss $< 0.5$, as opposed to 10/43 ($\sim 23\%$) for motif+NMA – corresponding to a roughly 17x-fold enrichment, while achieving comparable motif scaffolding performance (c.f. Fig. 11). However, much of this benefit appears to disappear in the process in inverse-folding and the subsequent re-folding. Joint motif+NMA conditioning still increases the relative chance of obtaining a sample with a low scNMA-score ($3/43 \sim 7\%$ of samples below 0.5) as compared to motif-only conditioning ($3/150 \sim 2\%$ below 0.5), roughly 3-fold, but the difference is much less pronounced than for the NMA-loss of the designed backbone (original NMA-loss). Figure 13 shows the scNMA-loss distribution for the motif+NMA and motif-only ESMFold designs. The best sample with low original NMA-loss is therefore not guaranteed to have similarly low scNMA-score. The pipeline inverse-folding and re-folding has also a surprising effect on the motif-only samples. Samples with high values of original NMA-loss are occasionally corrected to better NMA scores in the pipeline and match the targeted motif's dynamics better. Still, the in-

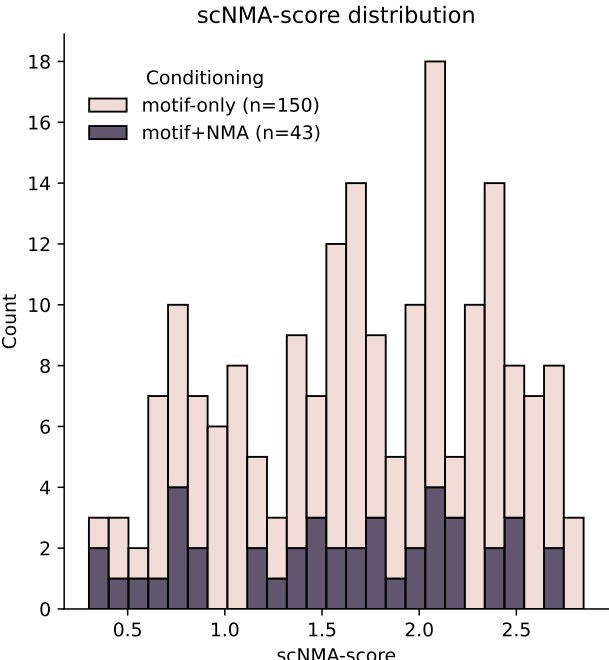

Figure 13: Distribution of the self-consistency NMA score (scNMA-score) of the backbone samples against structures obtained from inferring a sequence for each backbone via ProteinMPNN and refolding it via ESMFold. Joint motif+NMA conditioning increases the relative chance of obtaining a sample with a low scNMA-score ($3/43 \sim 7\%$ of samples below $0.5$) as compared to motif-only conditioning ($3/150 \sim 2\%$ below $0.5$) by roughly 3-fold, however the difference is much less pronounced than for the original NMA-loss. Again, the bars were stacked to avoid overlapping bars from being invisible.

troduction of the dynamics conditioning increases the relative chance of obtaining a sample with a low scNMA-score as compared to motif-only sampling. We leave the interesting question of how to retain high NMA-scores through inverse folding and re-folding pipelines as an interesting future work.

Lastly, we investigate how the scNMA-score correlates with the scMOTIF-RMSD. While the region where scMOTIF-RMSD<1Å remains unachievable for both motif+NMA and motif-only samples as previously discussed, the best samples (scMOTIF-RMSD and scNMA as low as possible) from all samples taken belong to the dynamics-conditioned subset.

## H.4 ADDITIONAL ALPHAFOLD2 DESIGNS

To give a visual intuition of the scores introduced above, we show the AlphaFold2 (AF2) designs of the Genie backbones - one motif-only conditioned, and one motif+NMA conditioned. Those backbones were deemed designable and close to designable by ESMFold - their scRMSD and pLDDT respectively were $1.923$ Å, $73.6$ for motif-only conditioned sample and $1.897$ Å, $68.1$ for motif+NMA conditioned sample. We repeated the inverse-folding and folding steps for these two selected samples with state-of-the-art AF2, and we computed the self-consistency scores again. The $C_\alpha$-only backbones derived from the AF2 designs are presented in the Figures 15 and 16. The displacement vectors in the lowest normal mode are attached to the points of the conditioned residues.

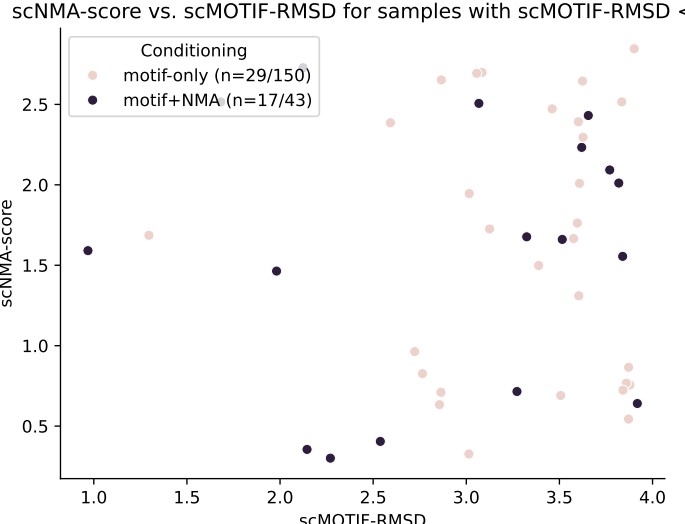

Figure 14: scNMA-score vs scMOTIF-RMSD for the motif+NMA and motif-only samples. We focus on samples with scMOTIF-RMSD $< 4$ as for structures with larger differences in the motif, the scNMA-scores likely become meaningless. As a consequence, 17 out of 43 and 29 out of 150 motif+NMA and motif-only samples are shown - the remaining are the outliers in the region scMOTIF-RMSD$> 4$Å.

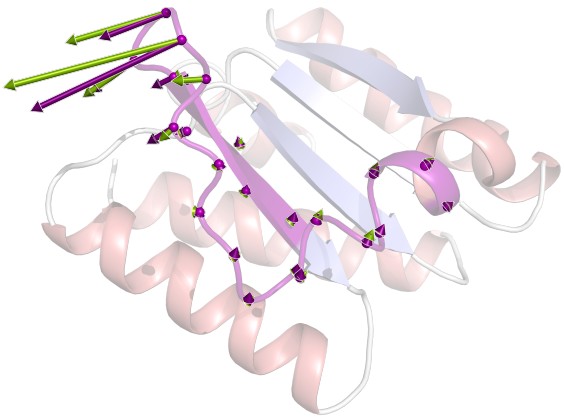

Figure 15: Backbone of an AF2 re-folded sequence obtained via ProteinMPNN from a motif+NMA conditioned raw backbone. Purple arrows are the displacements of the conditioned residues in the current structure, purple are the target rotated to the motif's frame of reference. Arrows are scaled up for visual clarity. Scores obtained with folding with AF2: scRMSD$= 1.49$, pLDDT$= 80.6$, scMOTIF-RMSD$=2.30$. Original NMA-loss$= 0.087$, scNMA-score$=0.29$.

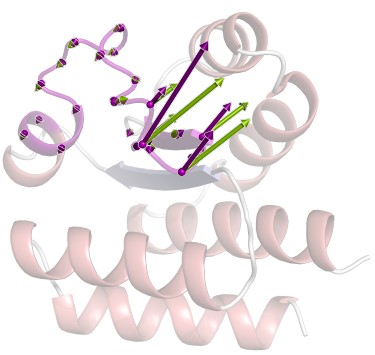

Figure 16: Backbone of an AF2 re-folded sequence obtained via ProteinMPNN from a motif-only conditioned raw backbone. Purple arrows are the displacements of the conditioned residues in the current structure, purple are the target rotated to the motif's frame of reference. Arrows are scaled up for visual clarity. Scores obtained with folding with AF2: scRMSD= 1.52, pLDDT= 81.6, scMOTIF-RMSD=2.07. Original NMA-loss=1.88, scNMA-score= 0.64.

