# OpenReview forum: "Dynamics-Informed Protein Design with Structure Conditioning"
_ICLR.cc/2024/Conference — ICLR 2024 poster_

### Official Review · Reviewer_65X8 · 2023-10-30

**Soundness:** 3 good
**Presentation:** 3 good
**Contribution:** 2 fair
**Rating:** 5
**Confidence:** 4

**Summary:**

The paper introduces a method for sampling protein structures with desired dynamic properties. This method builds on prior work in protein structure diffusions and computational analysis of protein dynamics. The primary contribution is a novel loss function for dynamics guidance. This loss function using normal mode analysis (NMA) to extract target displacement vectors per residue and then uses a differentiable NMA implementation to guide sampling towards the structures with displacements close to the targets. The authors present 27 conditional samples, which display loss values considerably lower than unconditional samples and self-consistency numbers to suggest that the sampled structures are designable.

**Strengths:**

The method is clearly presented If given the details of how NMA is performed, for example in a public codebase, I feel like I could straightforwardly reproduce the described method and results. The initial results also seem promising. The objective is clearly being optimized and the validity checks suggest that the sampled structures are reasonable.

**Weaknesses:**

The methodological and experimental contributions are both relatively minor.

Methodology:

The method combines classifier guidance on a Gaussian diffusion (Ho et al. 2020) with a NMA-derived loss function. Section 3.1 could be included in Section 2, as it is established and common to use a delta function centered at the denoiser's mean to perform guidance, and it is not unique to the paper. There also doesn't appear to be anything novel about the approach to NMA besides porting numpy code to pytorch. The contribution is the synthesis of NMA with classifier guidance, and the addition of a structure guidance term.

Experimental:

Novel methodology is not necessary for a significant contribution when a simple method can be convincingly shown to work well. This paper could be in this category, but I think the current results are fairly limited. Showing that guidance leads to lower loss values is more of a sanity check than a full result. Ideally the authors would demonstrate that the samples have the desired dynamics with a compelling independent evaluation. In this case, such an evaluation is obviously a challenging task, as it would require something like expensive molecular dynamics simulations or actual lab work, but establishing a compelling evaluation framework could be a very impactful contribution in itself. Likewise, it's a bit hard to know what scTM > 0.5 actually signifies, even though it has been used in prior work. Ultimately these structures would have to expressed as sequences. Are there compelling samples for those sequences such that these protein hinges could actually be created? It might be helpful to show that (1) the guidance method generalizes to the outcomes we really care about, not simply the proxy objective (2) there are high likelihood sequences that can (a) be expressed (b) have the desired dynamics.

**Questions:**

- How were the guidance scale values selected? Were they tuned for the protein structures you evaluated on (in which case they might be overfit to the tasks)? An independent evaluation model could also be useful here.

- Why is guidance only turned on in the middle of sampling? This is not standard practice to my knowledge.

---

> ### Author Response · Authors · 2023-11-18
> **Initial Response to Reviewer 4**
>
> Dear Reviewer 65X8,
>
> Thank you for your comments and suggestions on how to improve the paper.
>
> We hope that we address your comment about the novelty in the overall response. Briefly, we agree with you that our approach utilises the existing guidance framework (reconstruction based guidance). Yet we would like to highlight the simplicity of the method as an advantage rather than a drawback of our work. We would also like to emphasise that this particular choice of guidance strategy and schedule is not arbitrary, but was carefully chosen to match the strengths of NMA: In contrast to classifier-based guidance, reconstruction guidance results in more “protein-like” starting points for NMA analysis, for which it is more applicable. This is manifested by predicting the current estimate of the denoised position $\hat{x}_0$ via Tweedie’s formula before applying NMA instead of using the more noisy $x_t$, which would be used in standard classifier-based guidance and would require training an additional network to predict dynamics based on noisy version of the sample. We further tuned the guidance schedule to start only after a “burn-in” period in the sampling trajectory, after which a sufficiently protein-like structure is formed.
>
> We also discuss the points about evaluation in the general response, since they were of interest to multiple reviewers. For a short summary here, in addition to visual inspections and evaluating the NMA loss, the current quantitative proxy evaluation you mention, we newly add a self-consistent NMA (scNMA - explained in the general response) evaluation to our evaluation pipeline and are currently running experiments to assess it. Akin to the self-consistency metrics which are used in structure conditioning, this metric will evaluate how well the targeted mode is captured by the inverse folded & re-structure-predicted protein. We agree that evaluations which include molecular dynamics simulations will be a further, more powerful test that are out of scope for this current work, but that we plan to perform in the future.
>
>
> Please find here the answers to your other questions.
>
> 1. The guidance scales were indeed fine-tuned per target type, as is also common elsewhere in the field [1,2]. The same guidance scales were used for all samples with strain targets (when multiple sampled proteins had varying lengths), the same for all random targets (again, multiple sampled proteins with varying lengths), and there were different guidance scales for each of the hinge targets (per each target, many proteins, same lengths). Please find references to the works showing that fine-tuning guidance scale is a common practice in the field.
>
> 2. Why is the guidance scale tuned on in the middle of sampling?
> In our loss-guided diffusion, in the sampling at step t>0 we rely on a reasonably good prediction of sample $\hat{x}_0$ at t=0 via Tweedie’s formula. This is important to be in the realm of applicability of NMA, which is robust to slight changes in coordinates but not backbone topology. Early in the sampling the model estimate of the sample at t=0 is poor and fluctuates strongly such that starting conditioning too early has a negative impact on sample quality. We therefore tune the guidance scale on midway through the sampling process, when the sample has become more protein-like. We note that similar techniques (i.e. timestep dependent losses) are used by other works in the field, such as in FrameDiff [https://arxiv.org/abs/2302.02277], where the auxiliary loss on the atoms distances was switched on only when t<0.5*T_max.
>
> [1] Dhariwal and Nichol, “Diffusion Models Beat GANs on Image Synthesis”,  https://arxiv.org/pdf/2105.05233v4.pdf
>
> [2] Chung et al. , “Improving Diffusion Models for Inverse Problems using Manifold Constraints”, https://arxiv.org/pdf/2206.00941.pdf

---

> > ### Comment · Reviewer_65X8 · 2023-11-22
> >
> > Thank you for your response. It's worth pointing out that guidance using Tweedie's formula is also widely used and does not provide novelty either [1]. My persistent concern is that it is possible to find inputs that are essentially adversarial examples for your objective model $\log p(y | x_t)$, and you need some form of independent verification that samples with high values of the objective model are actually desirable. In image generation, visual inspection is sufficient for this purpose, because the appearance of the image is typically the objective, but it's not clear to me that visual inspection is sufficient in this case.
> >
> > At this time, I intend to keep my score unchanged.
> >
> > [1] Chung, Hyungjin, et al. "Improving diffusion models for inverse problems using manifold constraints." Advances in Neural Information Processing Systems 35 (2022): 25683-25696.

---

> > > ### Author Response · Authors · 2023-11-22
> > >
> > > Dear Reviewer 65X8,
> > >
> > > Thank you for taking the time to read and reply to our response.
> > >
> > > We apologise for an apparent miscommunication on our part: We did not mean to claim that guidance using Tweedie's formula is the novelty of our method (indeed, we cite reference [1] you mention in our response and the paper).
> > >
> > > Rather, we believe the novelty of our method comes from combining an old-school bioinformatics technique (NMA) and making it "machine-learning fit" and by formulating the task of joint motif and dynamics conditioning via normal modes. We believe that the introduction of this idea will be novel and interesting to several practicioners in the field, and hope that our evaluation of it constitute a helpful first guide.
> > >
> > > In the revised manuscript that we just uploaded (apologies for taking this long, re-running the experiments & analysis to address the interesting questions raised in the reviews took us some time), you will find an evaluation of how well improving the objective transfers to inverse-folded and re-folded designs exhibiting the targeted lowest normal mode dynamics. We agree that further independent analysis via techniques outside of NMA (e.g. molecular dynamics or lab experiments) are desirable to establish more clearly for which problems this technique an be promising and see these as important avenues of further work. For now, we rely on the established body of literature studying normal mode analysis and which types of motion it is useful for (e.g. larger scale backbone movements such as hinge or shear motions) to give an intuition about the realm where our ideas are likely to be applicable and useful. We agree that visual inspection does not constitute a sufficiently compelling argument and hope you find the self-consistency NMA score we introduce in this revised version of the manuscript (appendix G) helpful as criterion in this regard.
> > >
> > > We'd like to thank you again for your comments and for engaging in discussion, and hope you find the revised manuscript interesting.
> > >
> > > Kind regards,
> > > the authors

---

### Official Review · Reviewer_E2Vz · 2023-10-31

**Soundness:** 2 fair
**Presentation:** 2 fair
**Contribution:** 2 fair
**Rating:** 5
**Confidence:** 3

**Summary:**

The authors adapt a protein diffusion model to capture protein movement with an adaptation of normal model analysis.

**Strengths:**

I think it is good there is a clear biological system to focus on: movement, especially hinges.

The background on diffusion-based models was thorough and helpful.

I also think transferring the normal mode analysis to a tractable invariant loss is interesting and a good extension of current methods. For now, I’ve seen many invariant losses based on internal coordinates, but this is good.

**Weaknesses:**

Generally, I think this paper has very weak evidence to support the method works. Much of the analysis is based of heuristics or "by eye", which is not sufficient. If this ambiguity is expected, I anticipate the authors should generate methods, statistics, experiments, or simulations that can further support or reject the proposed hypotheses in this paper.

In detail:

Figure 1 is difficult to follow.
- What are the proteins in the lower row? They don’t seem to be the same proteins as those in the respective column as the top row. Is it the same set of motifs?
- If the arrows aren’t to scale to the entire protein, what are they to scale to?

Figure 2 - What should I be looking for? What is the range of good or bad numbers? I would strongly prefer if the dynamics of some other calibrated system could be shown in these plots, as the units and density are quite difficult to interpret.

Figure 3 - I’m not sure what I’m supposed to be looking for. What does a “bad” sample look like?

The filtering in section 5.1 is very ambiguous: “At the start, we filter out the ‘low quality’ samples whose mean chain distance is outside [3.75, 3.85]” What fraction were filtered out? “Occasionally the conditional sampling will ‘blow up’, which has been generally observed in many conditional diffusion models.” What does ‘blow up’ mean? Mathematically, can you define it, or is it all by eye?

**Questions:**

“We present the corresponding SDE theory as a formal justification of our approach.” In the abstract, what is SDE? Stochastic Differential Equations? If so, say that and then put SDE in ().

“Remarkably, Genie outperformed other models such as ProtDiff (Trippe et al., 2023), FoldingDiff (Wu et al., 2022) or FrameDiff (Yim et al., 2023), and remains comparable to RFDiffusion (Watson et al., 2022).” Why is this remarkable? Is it to the reader to determine if this is true? Should this evidence, however remarkable, be in an Appendix figure? Is RFDiffusion statistically significantly better?

“ the mean chain distance that should be close to 3.8 A ;” Where does 3.8 come from?

Do the values of NMA loss differ for strain or random targets? Do you expect them to? If so, this seems like a straightforward statistical quantity to calculate.

“In the analysis of the remaining samples, we considered the distributions of NMA-loss and scTM-score (Figure 5).” Isn’t the second figure RMSD? If they are the same, please define this.

---

> ### Author Response · Authors · 2023-11-18
> **Initial Response to Reviewer 3**
>
> Dear Reviewer E2Vz,
>
> Thank you for your valuable time and feedback, including points on figures 1, 2 & 3 – they help us improve the content and clarity of the manuscript.
>
> **On evaluation**:
> We address your concern on “weak evidence to support the method works” in the general response. Briefly, in addition to visual inspections and evaluating the NMA loss, the current quantitative evaluation, we newly add a self-consistent NMA (scNMA - explained in the general response) evaluation to our evaluation pipeline and are currently running experiments to assess it.
>
> **On your detailed points**:
> - “The filtering in section 5.1 is very ambiguous [...]”
> Apologies for any confusion caused. Out of 300 samples, for strain targets conditioned samples, 18 samples failed with NaNs, and 63 violated distance filter. Out of 300 samples, for random targets, 38 samples failed with NaNs, and for 62 samples violated distance filter. The unconditional samples never failed with NaN, but about 40-50 violated the chain distance filter. The NaN values occur due to what we previously referred to as “blow up”, explained in more detail in the next point.
>
> - “What does ‘blow up’ mean?”
> We apologise for the use of informal language and updated this statement in the manuscript. By ‘blow up’ we refer to a process where a coordinate increases by orders of magnitude along the sampling trajectory, or even becomes NaN. This phenomenon tends to happen when the conditioning pushes a sample’s coordinates outside of the realm of observed samples for which the denoiser was trained (and therefore it predicts mostly random scores there). This phenomenon underlines that finding a right balance between the NMA-loss driven part of the score and the unconditional part of the score requires striking a delicate balance, one of our contributions.
> This “divergence” effect is not unique to the denoiser we use and it has been generally found that many other large diffusion models working with image data suffer from diverging features in the sampling. In images they usually combat this with the ‘thresholding trick’ - they clip the feature values to never exceed some pre-defined range.
> Within the image domain that results in samples with too much colour saturation. For instance, in Stable Diffusion, which uses weights to balance conditional score and the unconditional score, too much weight on the conditional score leads to mode collapse and out of distribution samples [https://arxiv.org/pdf/2311.00938.pdf].
>
> - Q: “Remarkably, Genie outperformed other models such as [..] Why is this remarkable?” -  Apologies for the confusing phrasing. We meant for it to be read more like “Note that”. This is updated in the manuscript.
> We mention the capabilities of Genie model as compared to other models to explain our design choice to use Genie in the experiments. In this work we were not investigating how to improve the sample quality in the unconditional generation, but we needed to use a reasonable unconditional model to test the conditioning methodology. We chose Genie because of its satisfactory performance that according to the Genie paper [https://arxiv.org/abs/2301.12485] outperforms other models, as well as because of its code availability.
>
> - Q: “ the mean chain distance that should be close to 3.8 A  Where does 3.8 come from?” - thank you for this comment, we should have clarified that we refer to the distances of subsequent alpha carbons in the backbone here. The alpha carbon atoms (CA) in subsequent amino acid residues in a protein are physically constrained to lie in a very narrow range of around 3.8 Angstrom [c.f. e.g. https://www.wiley.com/en-us/Biochemistry,+4th+Edition-p-9780470570951]. Here we consider the mean $C_\alpha-C_\alpha$ distance, that is known to be close to 3.8 A.
>
> - Q: “Do the values of NMA loss differ for strain or random targets? Do you expect them to?” – Good question! As illustrated in Figure 2, the loss values are indeed fairly comparable between the two situations, however, we find it hard to give an intuition as to what to expect in these general cases. Instead, we would like to highlight that “strain” and “random” are meant mostly as more general demonstrations that the conditioning works for a variety of target definitions and substructures and that we use two kinds of targets in order to show that NMA-loss can be minimised regardless of how we choose the target. The “hinge” target formulation on the other hand is meant to capture the more application relevant situation of trying to design a certain motif with a targeted low mode behaviour.
>
> - Q: ““In the analysis of the remaining samples, we considered the distributions of NMA-loss and scTM-score (Figure 5). Isn’t the second figure RMSD?” - apologies for the confusing phrasing here. The distribution of NMA-loss is presented at Figure 5, scTM-score is discussed later in the text. We updated the phrasing in the text.

---

### Official Review · Reviewer_M3zP · 2023-11-01

**Soundness:** 3 good
**Presentation:** 3 good
**Contribution:** 3 good
**Rating:** 6
**Confidence:** 4

**Summary:**

The authors introduce a new problem for protein generative models of conditioning on the desired flexibility of parts of the protein. They develop a way to parameterize and model this flexibility and show that their conditioning can be applied on existing pre-trained models.

**Strengths:**

The authors introduce a previously unaddressed problem in protein generation of modeling structure flexibility and generating proteins with desired flexibility. Which biologically is a very important feature to model.

The introduced NMA loss is sensible and the results show that the proposed conditioning pipeline indeed works well.

**Weaknesses:**

While in general I liked the paper, one could say its a bit light on the content. The problem itself is novel and very interesting, but the solution is more or less the standard guidance framework with a new loss. Authors say that their approach could potentially be applied to other diffusion models, it would be nice to test at least a couple for this. Most validation is done by the same NMA loss that is used for guidance. So its unsurprising that guiding using a loss decreases it. So it would be nice to have more diverse validation metrics for the flexibility. Although admittedly I don't know what would be such better metrics. Maybe molecular dynamics literature has some suggestions?

**Questions:**

I don't have any further questions, but it would be nice if the authors could comment on the weaknesses outlined above.

A small note that in Section 3.1 you state that existing neural network approaches for fining matrix eigenvectors need re-training for each new matrix. In [1] there is a proposed architecture to generate sets of eigenvectors for graph Laplacians from a given distribution. While its a slightly different problem to what you were talking about, that architecture could potentially be used to generate eigenvectors for symmetric matrices without re-training as long as the training distribution matches the test matrices you feed.

[1] Martinkus, Karolis, et al. "Spectre: Spectral conditioning helps to overcome the expressivity limits of one-shot graph generators."

### After Rebuttal
I read all the reviews and author answers. While I find the overall problem interesting, I do tend to agree with the other reviewers that both the methodological novelty and rigorous evaluation is lacking. I will keep my score.

---

> ### Author Response · Authors · 2023-11-18
> **Initial Response to Reviewer 2**
>
> Dear Reviewer M3zP,
>
> Thank you for your comments and in particular the note on Section 3.1 with a suggested read. We now added the paper you recommend to the discussion of the design choices in Section 3.1.
>
> **Regarding your concern on the novelty of our method/content of the paper:**
> We agree with you that our approach utilises the existing guidance framework (reconstruction based guidance). Yet we would like to highlight the simplicity of the method as an advantage rather than a drawback of our work. We would also like to emphasise that this particular choice of guidance strategy and schedule is not arbitrary, but was carefully chosen to match the strengths of NMA: In contrast to classifier-based guidance, reconstruction guidance results in more “protein-like” starting points, for which NMA is more applicable. This is manifested by predicting the current estimate of the denoised position $\hat{x}_0$ via Tweedie’s formula before applying NMA instead of using the more noisy $x_t$, which would be used in standard classifier-based guidance. We further tuned the guidance schedule to start only after a “burn-in” period in the sampling trajectory, after which a sufficiently protein-like structure is formed.
>
> The problem of conditioning on dynamics has been widely discussed, but bringing the utilising and evaluating normal mode analysis as a conditioning target is a novel contribution of our work. Furthermore, we do not only propose this method but empirically demonstrate that we can generate samples that are still valid proteins but fulfil the targeted normal mode characteristics at the same time.
>
> Please find more discussion of these important points you raise also in the general response above.
>
>
> **Regarding your concern on the evaluation**:
> We agree with you that this work should be followed up by more in-depth evaluations via e.g. molecular dynamics simulations, however for this work we wanted to focus on establishing the principle behind our method. We are however currently running additional evaluation experiments to further support our claims (“scNMA” - self-consistency of NMA characteristics after inverse folding); for more information on this, please see the general response.

---

### Official Review · Reviewer_FtGW · 2023-11-01

**Soundness:** 2 fair
**Presentation:** 3 good
**Contribution:** 3 good
**Rating:** 6
**Confidence:** 4

**Summary:**

Tha manuscript presents a method for incorporating dynamics information in diffusion probabilistic models for protein generation. The central idea is to enforce the fluctuations in the samples to match the lowest mode of oscilation as predicted by a normal mode analysis. The authors demonstrate the general applicability of the method by combining it in a post-hoc fashion with the unconditioned Genie model to produce proteins compatible with provided dynamical properties.

**Strengths:**

To my knowledge, the presented method is novel. The manuscript is well written, and the paper appears technically sound. The conducted experiments are perhaps not as comprehensive as one might hope for, but they do demonstrate the basic premise of the approach. The problem addressed by this method is of fundamental importance, and any improvements in this area of research could therefore have significant impact.

**Weaknesses:**

While the paper reads well overall, there are parts where the clarity could be improved, in particular in the early description of the modelling task (what is being modelled), and in the technical description of the modeling approach (some variables undefined / details missing). The first point is in my view particularly important, because as it stands, it is difficult to read from the paper whether the method is generating: 1) protein sequence and structure, 2) protein backbone structure conditioned on sequence length, 3) protein structure conditioned on sequence, 4) protein backbone structure conditioned on nothing at all. It is to some extent possible to deduce these details from the conducted experiments, but I think this information should be stated more explicitly in the paper. Regarding technical details, these are more minor things that would make the paper easier to read. For details on both, see the questions below.

Related to the above, it would also benefit the paper if the authors briefly stated how they envision their approach to be useful in practice. If we have information about dynamics that we wish to impose on generated samples, we are presumably in a fairly constrained setting, where we wish to resample only parts of a protein. What are the advantages to conditioning a diffusion model on dynamics compared to for instance just using normal mode analysis (or coarse grained molecular dynamics) to generate a structural ensemble, and then use a model to predict amino acid identities conditioned on structure (i.e. some inverse folding model)?

Finally, the empirical evaluation of the method could have been stronger. Some of the results are primarily based on qualitative comparisons by-eye, or otherwise restricted to very specific motions. Clarification is also needed for some of the experimental setup (see details below).

**Questions:**

Page 3, eq (4)
As far as I could see, z has not been introduced. Is this just a sample from a unit normal? Please clarify.

Page 3, eq (6)
Isn't there an "x" missing on the right hand side?

Page 3. "Related work on Diffusion Probabilistic Models in protein context."
At this stage in the paper, after the general background on diffusion models, I would have expected a section on the specific modelling tasks that you intend to solve in this paper. Instead, you jump directly to related work, mentioning e.g. amino acid point clouds, which have not been defined. One solution could be move the related work to appear later in the paper, and go direclty to "3 Methods", but even that section does not give a complete description of what you are modelling (it introduces y_D, but not what x is, and how the amino acid identities appear in the model). For intance, if x is purely backbone structure, does that include all backbone atoms, or only C_alphas?

Page 3. "K ∈ R 3N×3Nis interaction constants matrix"
What is an interaction constants matrix? Is it derived from a force field? (EDIT: I see you have some info on this later, but would be good to clarify this when first mentioning NMA).

Page 4. "generate a new protein"
What exactly do you mean by "generating" a protein. Are we talking about both sequence and structure, conditioned on sequence length?

Page 4. eq (8)
Why is there no weighting factor (aka temperature) on the loss term? In other words, why would you expect this to be 1?

Page 5, "x_C_M \in ... is the expected positions of conditioned residues at t=0 as sampling progresses"
This sentence was confusing to me. Doesn't "sampling progresses" imply a "t" different from "t=0" (and at t=0, don't you know the position?). And in this case, shouldn’t this expected position be subscripted with a time t?

Page 6. "mean chain distance"
This is not clearly defined. From the reference value of 3.8, I assume this is the average CA-CA distance. Does this imply that you model the protein only at C_alpha resolution? If so, I would have expected this to be stated as part of the original model specification.

Page 7. "sample novel proteins using"
Again, are you sampling both structure and sequence here?

Page 7. "For each sample, we obtain 8 ProteinMPNN generated sequences with ESMFold (Lin et al., 2022) design for each"
This seems to suggest that "generating proteins" means only backbone coordinates, such that you need to fill in sequences using ProteinMPNN. But this sentence should probably be rewritten, since it is unclear what "with ESMFold (Lin et al., 2022) design for each" means. As far as I know ESMFold predicts structures, which is also consistent with the following sentences - but what does "design" then mean here? This should be clarified.

Page 7. "Occasionally the conditional sampling will ‘blow up’, which has been generally observed in many conditional diffusion models (Lou & Ermon, 2023)"
Does the rate of blowing up depend on the "guidance scale" used during sampling?

Page 7. "Figure 3 shows a pair of conditional and unconditional samples for one of the strain targets (additional sampled pairs are in Appendix E)"
Couldn’t this be quantified rather than verified through visual inspection on just a few targets?


### Minor comments:

Page 2, "as well as we perform the visual inspection."
Something is wrong in this sentence. Rephrase.

Figure 1, caption. "Bottom row: corresponding proteins synthesised with Genie"
What does the word “corresponding” imply? Are they conditioned on something similar?

Figure 1, caption. "Arrows are not to scale with respect to the entire protein"
Perhaps make this statement more precise by saying that arrows have been scaled up for increased visual clarity.

Page 3, "by the equivalence ∇x..."
This identify was not immediately apparent. I realize that it just requires a few steps , but for completeness it would be helpful if you could include the derivation of this in an appendix, or otherwise include a reference where this is done.

Page 5. "and expected displacements matrix v(x)"
I assume this means expected according to the NMA analysis, but perhaps state explicitly.

---

> ### Author Response · Authors · 2023-11-18
> **Initial response to Reviewer 1 (1/2)**
>
> Dear Reviewer FtGW,
>
> Thank you for thoroughly reviewing our manuscript and giving your insightful comments. We now improved the clarity of our manuscript according to your suggestions. Where suitable, we added pointers to the modifications in the manuscript that show where your concern has been addressed.
>
> Please also note the general points on **evaluation** and **novelty** that we discuss **in the general response**, since some questions were raised by several reviewers.
>
> Specifically to your point on **how we envision this method to be used**:
> If we have a selected motif (e.g. an enzymatic active site or pocket, binding cleft/pocket) that performs backbone rearrangements that are well captured by NMA during biological function (e.g. hinge or shear type backbone rearrangements), then this could be “grafted” onto a new scaffold that not only preserves the geometry of the motif, but also creates a scaffold backbone that promotes the relative motion of the motif that is needed to achieve the biological function. This idea is driven by the observation in the NMA community that some protein backbones pre-determine them for certain types of motion [https://pubmed.ncbi.nlm.nih.gov/16143512/], and that this motion is – for selected classes of proteins – the biologically relevant motion.
>
> We envision this to be useful to transfer functional motifs to different backbones which might have different additional binding domains to confer specificity, or lack thereof to promote promiscuity, while preserving the topology required for the functional motif’s primary motion.
> Besides coming from an existing protein, a functional motif with a dynamic behavior (e.g. a pincher motion) could also be the result of a protein engineering or design hypothesis.
>
> **Below is a point-to-point response to your further detailed comments**:
> (minor clarity/typo related points are not listed here, but addressed in the revised manuscript and highlighted with blue boxes)
>
>  - Page 4 eq (8) - in the current model specification, a weight factor in the loss function does not affect the final result
> Consider our loss in Equation 1 after we cast it to the form required in the conditional score (Equation 15) with an added temperature $\gamma$:
> $$
> \lambda_t \nabla  exp(-l(\hat{x}_0(x_t) ,y)/ \gamma) = - \frac{\lambda_t}{\gamma} \nabla  l(\hat{x}_0(x_t) ,y)
> $$
> with the guidance-scale $\lambda_t$. After reparametrisation $\hat{\lambda}_t  = \frac{\lambda_t}{\gamma}$
> $$
> \lambda_t \nabla  exp(-l(\hat{x}_0(x_t) ,y)/ \gamma) = - \hat{\lambda}_t \nabla  l(\hat{x}_0(x_t) ,y)
> $$
> the temperature $\gamma$ has no effect as $\hat{\lambda}_t$ is tuned.
>
> - Page 5, "x_C_M \in ... is the expected position of conditioned residues at t=0 as sampling progresses" - We rephrase that entire sentence in the updated manuscript, but please find additional clarification here.
> At each step in the sampling, which progresses from t=T_max to t=0, in order to calculate the NMA-loss or the structure loss we must obtain the model prediction of the residues of interest in the final backbone at t=0. By that we mean the “expected positions of conditioned residues at t=0“. Importantly, this is what differentiates the guidance paradigm we use here (reconstruction based guidance) from general classifier-based guidance – where a predictor network (or function) is needed that can predict using noisy data. While NMA is somewhat robust to noise, by predicting the current estimate $\hat{x}_0$ via Tweedie’s formula, we apply NMA to a more protein-like structure ($\hat{x}_0$ instead of $x_t$), moving us closer to the realm in which NMA is valid.

---

> > ### Author Response · Authors · 2023-11-18
> > **Initial response to Reviewer 1 (2/2)**
> >
> > - Page 7. "Occasionally the conditional sampling will ‘blow up’, which has been generally observed in many conditional diffusion models (Lou & Ermon, 2023)" - We apologise for the use of informal language and updated this statement in the manuscript, and your hypothesis that the “blow up” is connected to the guidance scale is indeed correct. By ‘blow up’ we refer to a process where a coordinate increases by orders of magnitude along the sampling trajectory, or even becomes NaN. This phenomenon tends to happen when the conditioning pushes a sample’s coordinates outside of the realm of observed samples for which the denoiser was trained (and therefore it predicts mostly random scores there). This phenomenon underlines that finding a right balance between the NMA-loss driven part of the score and the unconditional part of the score requires striking a delicate balance, one of our contributions.
> > This “divergence” effect is not unique to the denoiser we use and it has been generally found that many other large diffusion models working with image data suffer from diverging features in the sampling. In images they usually combat this with the ‘thresholding trick’ - they clip the feature values to never exceed some pre-defined range. Within the image domain that results in samples with too much colour saturation. For instance, in Stable Diffusion, which uses weights to balance conditional score and the unconditional score, too much weight on the conditional score leads to mode collapse and out of distribution samples [https://arxiv.org/pdf/2311.00938.pdf].
> >
> > - Page 2, "as well as we perform the visual inspection." Something is wrong in this sentence. Rephrase. - We rephrased this sentence for more clarity. We wanted to highlight that in addition to using our custom loss function for monitoring the effectiveness of our method, we also manually inspected the agreement of target and sample displacement vectors for selected samples and concluded that the results from our loss function agree with manual inspection. To further provide you with quantitative metrics, we are adding an experiment studying whether the designed-for normal mode is exhibited after inverse folding (we call this self-consistency NMA “scNMA” – refer to the general response for more details).
> >
> > - Figure 1, caption. "Arrows are not to scale with respect to the entire protein" Perhaps make this statement more precise by saying that arrows have been scaled up for increased visual clarity. - Addressed.

---

> > > ### Comment · Reviewer_FtGW · 2023-11-22
> > > **Response to rebuttal**
> > >
> > > Thanks to the authors for their rebuttal. The new manuscript introduces many small clarifications, which should make the paper easier to read. I still believe the paper would benefit from including more quantitative experiments, but I acknowledge that the paper is improved in its new version, and I will increase my score to 6.
> > >
> > > Minor comment: regarding the explanation of the 'blow up' phenomenon on page 7, the authors claim to have clarified this in the paper, but the manuscript seems to be unaltered in this section. I encourage the authors to check if they have uploaded the most recent version.

---

> > > > ### Author Response · Authors · 2023-11-22
> > > >
> > > > Dear Reviewer FtGW,
> > > >
> > > > Thank you for your response and for raising your score.
> > > >
> > > > We have just uploaded a revised manuscript which includes more quantitative experiments for an example that illustrates a biological application scenario (apologies for taking this long, running the experiments & extended analysis to address the interesting questions raised in the reviews took us some time). The revised version also contains the fix to your minor comment on the 'blow-up' discussion above.
> > > >
> > > > We hope you find the extra experiments and further quantitative evaluation in appendix G in the revised manuscript interesting, and would be delighted to hear your feedback if you find the time in the coming days.
> > > >
> > > > Kind regards,
> > > > the authors

---

### Author Response · Authors · 2023-11-18
**General response**

Dear Reviewers,

Thank you very much for your comments and thorough feedback. It helps us improve the quality of the paper.
We are pleased to hear that you find the **problem tackled here interesting**, the conditioning pipeline **clearly described** (RM3Pz, R65X8) and **technically sound** (RFtGW), and the **results promising** (R65X8). We are grateful for comments on how to improve the manuscripts content and clarity. The updated version of the manuscript contains the suggested clarifications. For your convenience, we **indicate changes** in the manuscript in **blue**, together with a blue box to signal which Reviewer’s points were addressed by the changes made.

Some of the Reviewers have raised similar points, therefore we address those in the overall response. When Reviewers have more specific concerns, we address them in the individual responses.


---
### General points:
**1) On novelty**
A concern was raised that the paper does not contribute anything novel to the NMA theory or conditioning methodology.

While we agree that classifier and reconstruction based guidance existed before, and that Normal Mode Analysis is an established technique in computational biology, the two have not been combined before and the potential and merit of using Normal Mode Analysis has not been explored in the community.

Further, we would like to highlight the fact that guiding a generation process with the NMA-loss is not guaranteed to result in high quality samples with the desired dynamics. This is an essential part of our work: we verify that this is indeed the case. The NMA-loss is only aware of the dynamical properties of the system and might guide the process to the structures that satisfy the dynamics constraints, but are not physically valid proteins. In the reverse diffusion, that is in the sampling process, the update step is driven by two competing forces - the unconditional score and the NMA-loss term. The two terms $\nabla_{x_t} \ln p(y | x_t) $ and $\nabla_{x_t} \ln p_t(x_t)$ might work together to generate a high quality sample, or might result in two gradient updates in completely opposite directions, and might thus fail to generate a high quality sample.

In our set of experiments with a varied set of targets where we generate a large number of novel samples and analyse those using some population statistics, we showcase that our proposed NMA-loss is indeed suitable for structure generation via a protein diffusion model. Through fine-tuning of the guidance scales in our experiments, we show that the desired balance between $\nabla_{x_t} \ln p(y | x_t) $ and $\nabla_{x_t} \ln p_t(x_t)$ is achievable and this is what we consider to be a major contribution of the paper.



**2) On evaluation**
It was also pointed out that the evaluation of the new samples is not comprehensive enough.

As Reviewer 65X8 noted, “the evaluation is obviously a challenging task, as it would require something like expensive molecular dynamics simulations or actual lab work”. The scope of this work was to introduce the NMA idea and investigate that it is possible to construct the dynamics-informed conditioning and that the proposed NMA-loss is something meaningful to optimise. The robustness of this conditioning method and its generalizability were the main focus of this work, therefore we mostly investigated the dynamics of the generated, coarse-grained protein backbones.  We agree that an extended evaluation of the generated backbones and of the conditioning pipeline via a molecular dynamics simulations is desirable, however due to the extensive scope of these evaluations we consider this part of future work.

To nonetheless improve evaluation, and to do justice to Reviewer 65X8’s comment highlighting the fact that “ultimately these structures would have to (be) expressed as sequences”, we include an additional “self-consistency NMA” check to the updated evaluation pipeline. This evaluation is akin to the “self-consistency” metrics (e.g. scRMSD, scTM) that are commonly used to evaluate the success of structural conditioning and measures whether an inverse-folded sequence designed from the targeted backbone exhibits the targeted NMA behaviour.

We are currently running additional experiments that include this evaluation step, specifically focussing on the biologically relevant scenario of *jointly conditioning on structure and dynamics*, to illustrate how our framework might be used in practice, as requested by Reviewer FtGW. We will post an update with the results here as soon as we can.

---

> ### Author Response · Authors · 2023-11-22
> **Updated experiments to answer rebuttal questions**
>
> Dear Reviewers,
>
> We have now updated the manuscript with the additional experiments in the Appendix G. We hope you will find them helpful in addressing any of your remaining questions.
>
>
> Please find a **quick summary** below:
> - The additional experiments **extend our evaluation framework with a self-consistency NMA score (scNMA)**. This metric quantifies whether the structure obtained from the pipeline: Genie $C_\alpha$ backbone -> ProteinMPNN inverse-folding -> ESMFold folding has the NMA-loss close to the value derived from the backbone.
> - In the conducted experiments we focused on a **target that illustrates a biologically relevant scenario**: a segment of 22 residues containing the lysozyme active site, that is involved in the lysozyme hinge motion. To evaluate the sample quality and the method’s success, we used the previously described scRMSD, scTM, pLDDT and pAE criteria.
> - We found that for the chosen target, the structure conditioning problem is a challenging task itself. Motif-only conditioned samples had low designability scores. Joint motif+NMA conditioning surprisingly improved the designability score slightly
> - The extra evaluation **confirms that indeed our methodology allows for generating raw $C_\alpha$ backbones with the targeted dynamics, which we believe is a major contribution of our work**: Only 2/150 ($\sim1.3$%) of motif-only backbones achieve $\text{NMA-loss} < 0.5$, as opposed to 10/43 ($\sim23$%) for motif+NMA -- corresponding to a **roughly 17x-fold enrichment**, while achieving comparable motif scaffolding performance (c.f. Fig. 11 & 12).
> - However, much of this benefit seems to disappear in the process of inverse-folding and the subsequent re-folding. After inverse-folding and re-folding, **joint motif+NMA conditioning** still **increases the relative chance** of obtaining a sample with a **low scNMA-score** ($3/43 \sim 7$% of samples below $0.5$) as compared to motif-only conditioning ($3/150 \sim 2$% below $0.5$), **roughly 3-fold**, but the difference is much less pronounced than for the NMA-loss of the designed backbone (original NMA-loss).
>
>
> We would like to again highlight the contribution of our work which is a methodology for dynamics-informed backbone design. This is a first work aiming at directly integrating dynamics information to the protein diffusion model and we hope it will be of benefit to a wider scientific community.
>
> We hope you find the new analysis satisfactory and encouraging for the score reconsideration. Thank you for your constructive feedback that has been invaluable in advancing our evaluation pipeline.

---

> > ### Author Response · Authors · 2023-11-23
> >
> > Dear Reviewers,
> >
> > Please note we added a minor correction to the manuscript regarding the usage of ProteinMPNN in the Appendix G.

---

### Meta-Review · Area_Chair_VQSu · 2023-12-05

**Metareview:**

This manuscript introduces a method that integrates dynamic information into protein generative models, focusing on conditioning diffusion probabilistic models on protein dynamics, especially on the lowest non-trivial normal mode of oscillation. The approach employs a novel loss function based on Normal Mode Analysis (NMA) to guide the generation of protein structures to meet specific dynamic properties. The practical utility of this method is demonstrated by adapting the existing Genie model to produce proteins that meet both structural and dynamic criteria, without requiring retraining.

**Justification For Why Not Higher Score:**

While the work is noted for its interesting application and practical significance, it is critiqued for its limited technical innovation. The experimental results could also have been more extensive.

**Justification For Why Not Lower Score:**

The specific type of dynamic conditioning presented in this study has not been previously considered, adding a unique aspect to the field of protein design by incorporating dynamic behaviors into protein structures.

---

### Decision · Program_Chairs · 2024-01-16

Accept (poster)